# CryoEM structures of anion exchanger 1 capture multiple states of inward- and outward-facing conformations

Hristina R. Zhekova[1,6], Jiansen Jiang [2,3,4,6], Weiguang Wang[2,3,4,6], Kirill Tsirulnikov[2], Gülru Kayık[1], Hanif Muhammad Khan[1], Rustam Azimov[2], Natalia Abuladze[2], Liyo Kao[2], Debbie Newman[2], Sergei Yu. Noskov [1], D. Peter Tieleman [1], Z. Hong Zhou [3,4], Alexander Pushkin[2] & Ira Kurtz [2,5✉]

Anion exchanger 1 (AE1, band 3) is a major membrane protein of red blood cells and plays a key role in acid-base homeostasis, urine acidification, red blood cell shape regulation, and removal of carbon dioxide during respiration. Though structures of the transmembrane domain (TMD) of three SLC4 transporters, including AE1, have been resolved previously in their outward-facing (OF) state, no mammalian SLC4 structure has been reported in the inward-facing (IF) conformation. Here we present the cryoEM structures of full-length bovine AE1 with its TMD captured in both IF and OF conformations. Remarkably, both IF-IF homodimers and IF-OF heterodimers were detected. The IF structures feature downward movement in the core domain with significant unexpected elongation of TM11. Molecular modeling and structure guided mutagenesis confirmed the functional significance of residues involved in TM11 elongation. Our data provide direct evidence for an elevator-like mechanism of ion transport by an SLC4 family member.

[1] Centre for Molecular Simulation, Department of Biological Sciences, University of Calgary, Calgary, Canada. [2] Department of Medicine, Division of Nephrology, David Geffen School of Medicine, University of California, Los Angeles, CA, USA. [3] Department of Microbiology, Immunology and Molecular Genetics, University of California, Los Angeles (UCLA), Los Angeles, CA, USA. [4] California NanoSystems Institute, UCLA, Los Angeles, CA, USA. [5] Brain Research Institute, University of California, Los Angeles, CA, USA. [6] These authors contributed equally: Hristina R. Zhekova, Jiansen Jiang, Weiguang Wang. ✉email: ikurtz@mednet.ucla.edu

AE1, a member of the solute carrier ion transport family 4 (SLC4), mediates the exchange of $HCO_3^-$ with $Cl^-$ across plasma membranes[1,2]. AE1 (SLC4A1) constitutes ~30% of the membrane mass of the red blood cell, amounting to ~$10^6$ AE1 molecules per cell[1,2]. Carbon dioxide generated by metabolic processes in the tissues diffuses into the red blood cells where it reacts with water to form $HCO_3^-$ and protons in a reversible process catalyzed by carbonic anhydrase II[1,2]. As the $HCO_3^-$ concentration in the erythrocyte increases, AE1 mediates the electroneutral exchange of $Cl^-$ for $HCO_3^-$ at the extremely fast exchange rate of ~50,000 ions per second[1–3]. Thus, ~90% of the $CO_2$ is taken from the tissues to the lungs as the more soluble form, $HCO_3^-$. In the lungs, the process is reversed and $CO_2$ is exhaled[1,2,4]. AE1 also plays an essential role in the acidification of the urine in the collecting duct[5]. Inherited mutations in AE1 cause anemia with red cell morphological abnormalities as well as distal renal tubular acidosis[1,2,6].

Mammalian erythrocyte AE1 is a ~100 kD integral trans-membrane glycoprotein consisting of an N-terminal cytoplasmic domain (CD, residues 1 to 360 in human AE1) and a C-terminal trans-membrane domain (TMD, residues 361 to 911 in human AE1)[1,2]. The CD plays an important role in cytoskeletal attachment and binding to various proteins, e.g., ankyrin, band 4.2 and band 4.1 proteins, glycolytic enzymes, hemoglobin, deoxyhemoglobin, hemichromes, p72syk protein tyrosine kinase, glycophorin A, adducin and integrin-linked kinase, and is proposed to play a key role in mediating red blood cell elasticity[1,2,7,8]. The TMD, which mediates anion exchange[9,10], has a $7 + 7$ inverted repeat fold and is subdivided into core and gate domains similar to the human electrogenic sodium bicarbonate cotransporter 1 (NBCe1, SLC4A4)[11], the rat sodium dependent chloride carbonate exchanger (NDCBE, Slc4a8)[12], the plant boron transporter Bor1[13] homologous to the mammalian $H^+/NH_3$ transporter SLC4A11, the fungus purine symporter UapA[14], the bacterial uracil transporter UraA[15], the bacterial proton-coupled fumarate symporter SLC26Dg[16], and the murine $Cl^-$ transporter Slc26a9[17]. The crystal structures of the human AE1 CD expressed in bacteria, and the TMD in the OF conformation purified from human erythrocyte membranes, as well as low-resolution negative stain EM structures of full-length mammalian AE1 were previously reported[7–9]. Preliminary cryoEM full-length human AE1 structures in apo-, and holo-forms with substrates/inhibitors reported TMD structures solved in the OF conformation[18]. The structures of the membrane domain of other mammalian SLC4 transporters, NBCe1[11] and NDCBE[12], were also resolved only in the OF state. Thus far, no $7 + 7$ inverted repeat fold transporter has been resolved in both IF and OF state precluding the analysis of their underlying transport mechanisms. Given that AE1, one of the fastest ion exchangers[1,19], is of key importance in mammalian biology and general studies of secondary transport by proteins, understanding of its transport mechanism is a priority and requires resolution of its IF structure.

Here, by single-particle cryoEM, we determined the structures of the TMD of bovine AE1 at near-atomic resolution and captured both IF and OF conformations. Heterodimers (OF–IF) of bovine AE1 were detected in addition to homodimers. The IF state features an elongated TM11, which incorporates residues from intracellular loop 5 (IL5) between TMs 10 and 11, and partially unfolded TM10. The most significant conformational change occurs in the core domain, which combines a downward movement with small lateral displacement and a slight rotation respective to the gate domain in the OF to IF transition. The data supports an elevator-like transport mechanism combined with rotational movement of the ion coordination site and reorganization of IL5 between β-hairpin and α-helical form as it incorporates into TM11. Computational modeling demonstrates that the IF cavity is well hydrated and that anions tend to accumulate at the interface between the two monomers at the intracellular side of AE1 from where they can access laterally the IF cavity.

## Results and discussion

**Full-length AE1 structures demonstrate existence of various conformational states in bovine AE1 TMD.** AE1 purified from bovine red blood cell according to our method[8], was ~98% purity and predominantly dimeric (Supplementary Fig. 1a, b). Dose-fractionated frames were recorded as movies by direct electron-counting and subsequently aligned to produce averaged cryoEM micrographs, which show well dispersed full-length AE1 particles (Fig. 1, Supplementary Fig. 1c). Movie recording with electron-counting detection technology[20] and frame alignment with the dose-weighting[21], together allowed for the visualization and subsequent 2D image classification of this ~100 kDa membrane protein. The 2D class averages (Supplementary Fig. 1d) reveal various views of particles with discernable TMD and CD as well as obvious features of a dimeric configuration in both the TMD and the CD. The dimerization of full-length AE1 is consistent with the reported structures of the TMD of human AE1[9], human NBCe1[11] and rat NDCBE[12]. In some 2D class averages, the cryoEM density of the CD is fuzzy or invisible, suggesting a high mobility of the CD and a flexible linkage between the CD and the TMD in agreement with our previous low-resolution EM study[8].

3D classification using the C1 symmetry resulted in three major classes that differ mostly in the position of the CD (Fig. 1a–c, Supplementary Fig. 1d–g). Two classes showed fully connected CD and partially connected CD respectively. 3D refinement of these two classes yielded 3D reconstructions at a resolution of 6.3 Å and 6.9 Å respectively (Table 1). One class has no CD. The class with partially connected CD was found to have one monomer in IF state

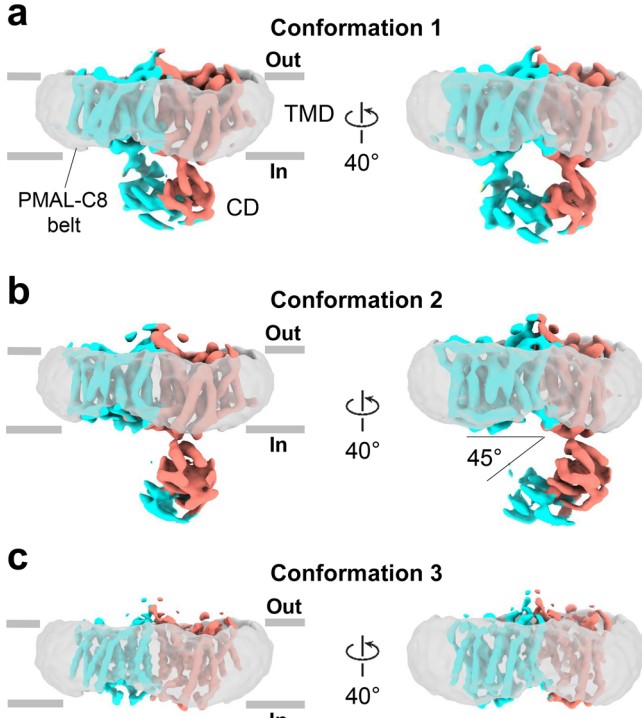

**Fig. 1 Structures of the full-length AE1 showing multiple conformations. a** CryoEM map of the IF–IF state with fully connected CD. **b** CryoEM map of the IF–OF state with partially connected CD. **c** The final IF–IF reconstruction of the TM region. The two monomers are colored in light blue and salmon respectively. The amphipol-C8 belts are shown as a transparent gray surface.

**Table 1 Cryo-EM data collection, refinement, and validation statistics.**

|  | (IF-IF)$_{TMD}$ EMDB-27267 PDB 8D9N | (IF-IF)$_{Full-length}$ EMDB-28055 PDB 8EEQ | (IF-OF)$_{Full-length}$ EMDB-27856 PDB 8E34 |
|---|---|---|---|
| **Data collection and processing** |  |  |  |
| Magnification | 36,764 | 36,764 | 36,764 |
| Voltage (kV) | 300 | 300 | 300 |
| Electron exposure (e⁻/Å²) | 52 | 52 | 52 |
| Defocus range (μm) | −1.4 to −3.2 | −1.4 to −3.2 | −1.4 to −3.2 |
| Pixel size (Å) | 1.36 | 1.36 | 1.36 |
| Symmetry imposed | C2 | C1 | C1 |
| Initial particle images (no.) | 2,635,578 | 2,635,578 | 2,635,578 |
| Final particle images (no.) | 251,871 | 90,818 | 87,780 |
| Map resolution (Å) | 4.4 | 6.3 | 6.9 |
| FSC threshold | 0.143 | 0.143 | 0.143 |
| **Refinement** |  |  |  |
| Initial model used (PDB code) | 4YZF | 4YZF, 4KY9 | 4YZF |
| Model resolution (Å) | 4.5 | 6.5 | 7.1 |
| FSC threshold | 0.5 | 0.5 | 0.5 |
| Map sharpening B factor (Å²) | −275.87 | −283.70 | −233.43 |
| **Model composition** |  |  |  |
| Non-hydrogen atoms | 7,188 | 11,516 | 7,553 |
| Protein residues | 902 | 1,446 | 948 |
| Ligands | 0 | 0 | 0 |
| **R.m.s. deviations** |  |  |  |
| Bond lengths (Å) | 0.003 | 0.005 | 0.003 |
| Bond angles (°) | 0.594 | 1.105 | 0.714 |
| **Validation** |  |  |  |
| MolProbity score | 1.78 | 2.01 | 1.97 |
| Clashscore | 3 | 5.70 | 12.02 |
| Poor rotamers (%) | 0 | 0.49 | 0 |
| **Ramachandran plot** |  |  |  |
| Favored (%) | 94.16 | 94.61 | 94.56 |
| Allowed (%) | 5.84 | 5.39 | 5.33 |
| Disallowed (%) | 0 | 0 | 0.11 |

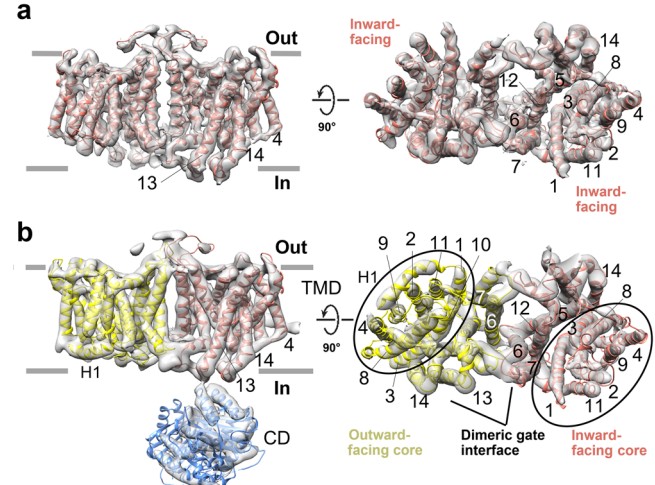

**Fig. 2 Coexistence of IF and OF monomers in AE1 dimer. a** Bovine AE1 dimer with both monomers in the IF state. **b** Bovine AE1 dimer with monomers in OF (yellow) and IF (salmon) conformations. The CD and OF and IF cores, as well as the dimeric gate interface are also shown.

SLC4 structures have been resolved as OF–OF dimers[9,11,12]. Here, we have resolved not only IF–IF homodimers, but also heterodimers (OF–IF) of bovine AE1 (Fig. 2). The finding that monomers within an SLC4 dimer can have different conformations, suggests that they can operate independently from one another, in agreement with previous studies by Macara and Cantley[26].

**The IF and OF states of the TMD of AE1.** In the cryoEM 3D reconstructions, the TMD appears more stable than the highly mobile CD. The IF and OF states extracted from the bovine AE1 cryoEM maps feature the 7 + 7 TM inverted repeat fold architecture, observed in several SLC4, SLC26, and SLC23 proteins[11–14,16,19]. Namely, each TMD consists of 14 TMs, divided into two structurally related groups of inverted repeats (TMs 1–7 and TMs 8–14). The TMD can be separated into gate (TMs 5–7,12–14) and core domain (TMs 1–4, 8–11) (Fig. 2, Supplementary Fig. 2). In the AE1 dimer, the two gate domains form the dimerization interface at the dimer center and the two core domains are located at the opposite ends of the dimer. Comparison between the structure of human AE1 TMD, locked in the OF open state by H$_2$DIDS crosslinking between lysine residues in TMs 5 and 13, and our bovine AE1 TMD structures in the IF and OF conformations reveals a nearly identical conformation for their gate domains (Fig. 3, Supplementary Figs. 3 and 4). The structures of two other SLC4 transporters (NBCe1 and NDCBE) also show similar organization and dimerization of the gate domain[11,12], suggesting that homo-dimerization through the gate domain is conserved in the SLC4 family, and that the conformation of the gate domain remains stationary during anion translocation without notable movement in the membrane. The core domains of bovine and human OF AE1 demonstrate significant similarity of all involved transmembrane segments upon overlap of the structures (Supplementary Fig. 3). In contrast, the core domain shows a remarkable difference between the OF human and bovine AE1 on one hand and the IF bovine AE1 on the other hand (Fig. 3, Supplementary Fig. 4). This difference comes from rigid rotation of the core domain ~20° with respect to the gate domain followed by diagonal downward displacement of the core with <5 Å along the z axis and small lateral motion in the XY plane (Fig. 3, Supplementary Movie 1). The overall vertical motion of the ion coordinating residues in the core domain is ~5 Å. Further inspection of the IF structure of bovine AE1 TMD

and the other in OF state. The classes without CD and fully connected CD are both monomers in IF state. Particles from these two classes were combined and refined using a soft mask of the TM region. The final reconstruction of the IF state was resolved at 4.4 Å. (Supplementary Fig. 1e–g, Table 1). The overall size and shape of these reconstructions suggest unambiguously that they are dimers of full-length AE1, each with a PMAL-C8 buried TMD and an associated CD (Fig. 1a, b). Both the TMD and CD dimers have an elongated shape. Unlike in previously reported models, which proposed that the TMD and the CD are aligned in parallel to form a large buried surface area[22–24], the CD in some dimers is rotated as much as 45° pivoting around one of the CD-TMD contact points (Fig. 1a, b; Supplementary Fig. 1f, g) in the dimers, resulting in a more extended linker between the CD and TMD of the other monomer (Fig. 1b).

These data indicate that the linker between the CD and TMD monomers is not rigid and may be extended, agreeing with the results of our previous AE1 study[8]. It is currently unknown what the molecular forces causing these conformational changes in bovine AE1 are. The limited direct contact between the TMD and the CD leads to a large corridor between the two domains, providing an unobstructed access for the substrate ions to the intracellular side of the TMD. An ion permeation pathway in the CD, as suggested before[25], is not necessary for the transport function of the TMD in SLC4 transporters in agreement with studies indicating that removal of the CD does not have a major effect on the ion transport in AE1[1,2].

**Homo- and heterodimers of AE1.** The predominant oligomeric form of bovine AE1 is dimeric[1,2], and all previously reported

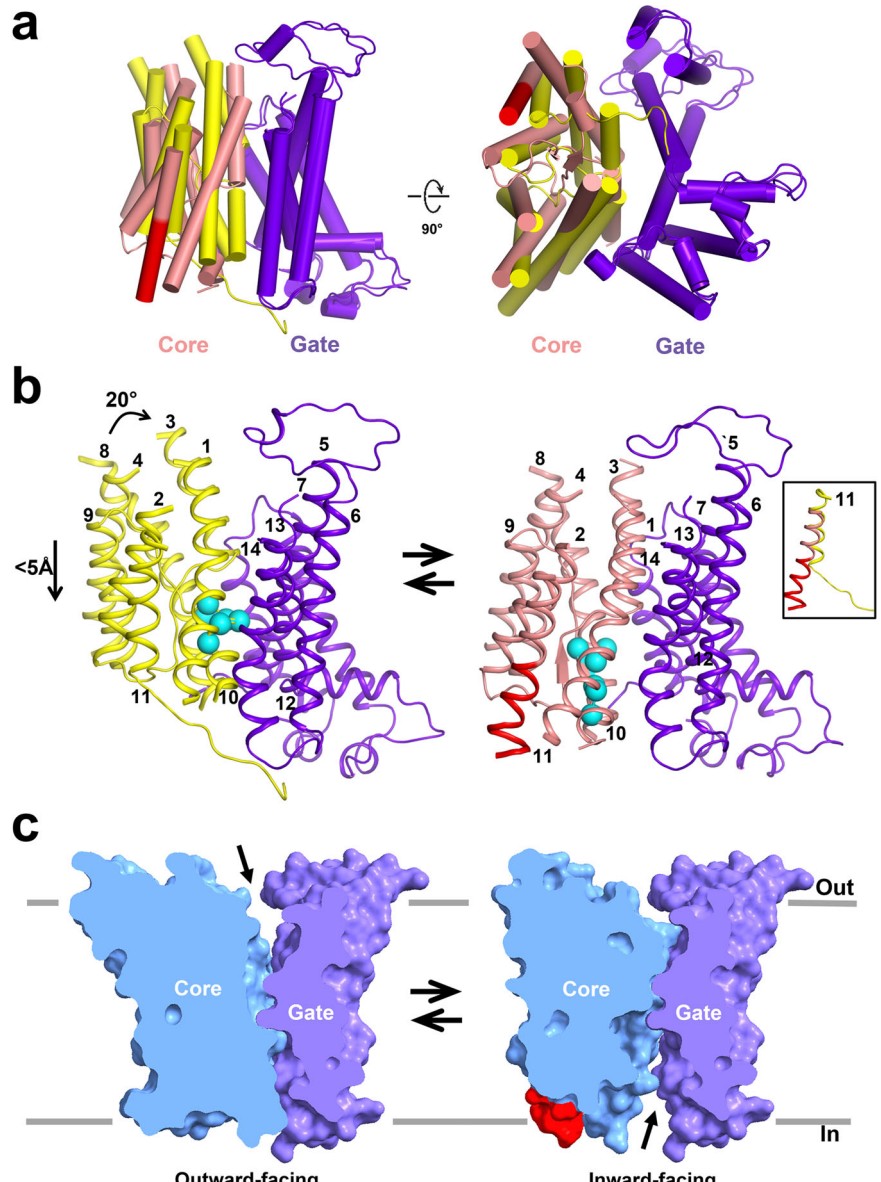

**Fig. 3 Comparison of OF and IF states of bovine AE1. a** Superposed atomic models of OF and IF states showing significant conformational rearrangements of the core domain. In the IF state, TM11 extends significantly (red cylinder). Gate domain (purple), IF core domain (salmon) and OF core domain (yellow). **b** Comparison of OF and IF states of AE1 demonstrating downward movement with some rotational motion of the core domain with respect to the gate domain during the OF to IF transition (see also Supplementary Movies 1–3). Cα atoms of previously identified core residues from the central S1 site[27] are shown as cyan spheres. The inset illustrates the difference in TM11 in the OF and IF state. **c** Surface models of the OF and IF structures. The OF and IF cavities are indicated with arrows.

reveals that, due to the rotation and downward motion of the core domain, the ion coordination site becomes open to the cytoplasm and fully closed to the extracellular side, i.e., it is in the IF open state not previously observed in AE1. The linkers connecting the core and gate domains, including H2 between TMs 4 and 5, the long extracellular loop 3 (EL3) between TMs 7 and 8, EL4, and H3 between TMs 11 and 12, appear flexible and do not hinder the movement of the core domain. Pronounced differences in the IF state are also evident in TM11 (elongated) and TM10 (partially unfolded) (Fig. 3, Supplementary Fig. 4, Supplementary Movies 2 and 3). IL5 between TM10 and TM11, which assumes an anti-parallel β-hairpin conformation in the OF state, loses this β-hairpin structure and part of it folds into the elongated α-helix of TM11 in the IF state (Fig. 3, Supplementary Fig. 4). The concerted movement of TMs 10 and 11 and the β-hairpin to α-helix

transition of IL5 opens a large cavity at the intracellular side and exposes residues R748 and E699 (analogous to residues R730 and E681 in human AE1) to the intracellular solution. These residues have been previously identified as part of the binding pocket in the OF conformation of human AE1[9]. At the extracellular side, the downward movement and rotation of TMs 1 and 3 occludes the large permeation cavity seen in the OF state and shields the binding pocket residues I546, F550, E699, and T745-S749 at the previously identified central ion binding site S1 of AE1[27] from the extracellular solution (Fig. 3). The IF conformation is stabilized by several salt bridges between the residues of TMs 1, 3, 5, and 13 (D447-K557, K448-E553, E491-R869) (Supplementary Fig. 5), which aid further in the occlusion of the OF cavity.

Supplementary movies 2 and 3 present the IF to OF transition of bAE1 calculated from the elastic network model driven Brownian

Dynamics simulations (eBDIMS) server[28]. In the eBDIMS method, principal components of conformational change are evaluated on the basis of provided IF and OF endpoints. The two largest principal components are then used for evolution of the conformational transition with coarse grained elastic network Brownian Dynamics simulations. The method does not require manual selection of collective variables and has been applied successfully in qualitative description of the conformational transitions in a number of challenging protein systems[29]. Upward and downward movements of the AE1 core with respect to the gate, consistent with elevator-like motion, are clearly visible from the generated eBDIMS trajectories. The concerted motion of TM10–IL5–TM11 and elongation of TM11, which lead to opening and occlusion of the IF permeation cavity of bAE1 are also shown in the movies. Similar occluding motion of TM10–IL5–TM11, starting from an IF open state occurs naturally in some of the 1 μs unbiased MD replicas (Supplementary movies 4 and 5).

**Ion dynamics in the IF and OF states**. AE1 exchanges chloride and bicarbonate anions in a sodium independent manner[1,2,30]. To study the dynamics of anions and cations in the permeation cavities formed in the IF and OF states, we performed Site Identification by Ligand Competitive Saturation (SILCS)[31] calculations of bAE1 monomers in IF and OF state and multiple 1 μs MD simulations of the symmetric (IF–IF) and asymmetric (IF–OF) dimers, embedded in a POPC bilayer and in the presence of a 0.75 M NaCl + 0.75 M NaHCO₃ solution. Figure 4a presents the preferred localization sites for anions (cyan mesh) and cations (yellow mesh) determined from the SILCS maps. Both IF and OF states of bAE1 feature large water accessible permeation cavities (Supplementary Fig. 6), which allow ions from the solution to diffuse to the protein center. The OF permeation cavity of bovine AE1 (Fig. 4a, left) is exclusively accessible for anions from the extracellular solution, which traverse freely the permeation cavity to the central R748 residue of sites S1 and S2, in line with previous SILCS studies on OF human AE1[27]. In the IF state, the anions from the intracellular solution accumulate at the entry of the permeation cavity (Fig. 4a, right) drawn by the cluster of positively charged residues (R607, K608, K610, K618, R620, and R621), from TMs 6 and 7, which form part of the gate domain at the dimer interface, and the central R748 residue, which in the IF state has traveled toward the IF permeation cavity propelled by the downward motion of TM10.

To assess further the dynamics of the physiological ions (Cl⁻ and HCO₃⁻) transported by AE1, we evaluated ion densities from 1 μs MD simulations of the bAE1 dimers. The bicarbonate densities in the asymmetric (IF–OF) dimers (cyan iso-surface) are mapped in Fig. 4b, Supplementary Fig. 7. The Cl⁻ maps indicate Cl⁻ presence in the same areas (Supplementary Fig. 7), although their density is significantly lower due to the more short-lived Cl⁻ occupancy, compared to HCO₃⁻. There is a marked difference in the ion permeation dynamics at both sides of the membrane. At the intracellular side, the ions tend to accumulate in the region of the positively charged residues (R607, K608, K610, K618, R620, and R621) in agreement with the SILCS maps (Fig. 4a, right). This region forms a deep anion reservoir at the dimer interface and is consistently occupied with multiple HCO₃⁻ and Cl⁻ ions during the MD trajectories. The anions can then move laterally from the reservoir toward the hydrated IF cavity and the partially unfolded TM10 and R748. Such accumulation of anions is not observed at the extracellular side of the dimer interface, due to the position of TMs 5 and 6 and the EL3 occupying this area. Instead, the anions permeate vertically from the extracellular solution toward the wide hydrated OF cavity (Fig. 4b, left) drawn by several positively charged residues (K557, R748, R869, K872) as described previously for human AE1[9,27].

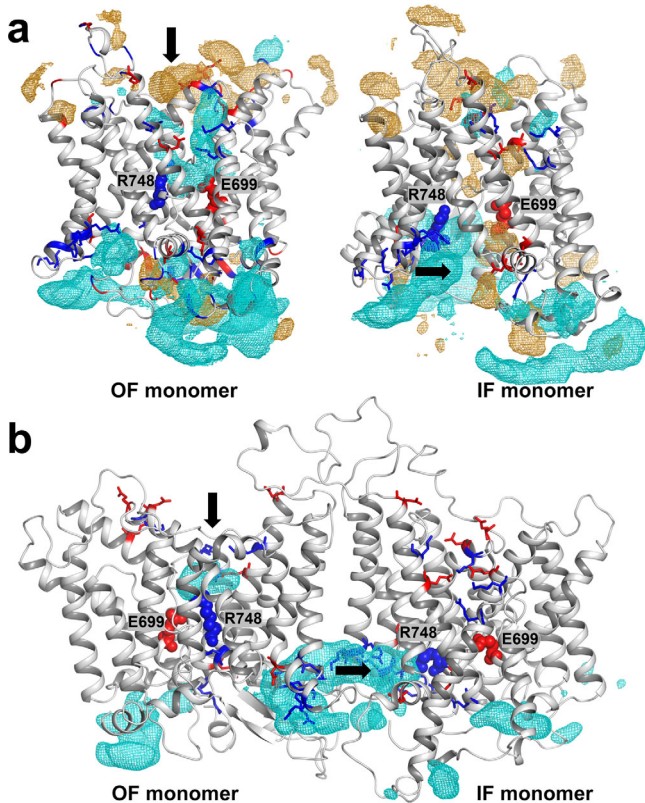

**Fig. 4 SILCS and MD simulations of AE1. a** Cation (orange mesh) and anion (cyan mesh) maps from SILCS simulations of AE1 monomers visualized at Grid Free Energy level of −0.9 kcal/mol. **b** Average HCO₃⁻ density maps (cyan mesh) calculated from 1 μs MD simulations of three IF–OF replicas (contour isovalue 0.05). Positively and negatively charged residues lining the IF and OF permeation cavities are shown as blue and red sticks, respectively. Residues R748 and E699 from the central site S1 in the OF state are shown as blue and red spheres, respectively. The ion entry pathways in the OF and IF cavities in the monomers and the dimer are presented as black arrows (vertical for ion permeation in the OF cavity and lateral for ion permeation in the IF cavity).

**Functional mutagenesis**. Mutagenesis studies of several residues of human AE1 from TM11 and IL5, the protein area involved in TM11 elongation and β-hairpin to α-helix transition, demonstrate the functional importance of this region (Fig. 5, Supplementary Fig. 8). Although significant decrease in Cl⁻-driven base flux to about 40–60% of AE1 activity was observed in cysteine mutants of M741, A744, P747, A750, A751, Q754, and L765 (corresponding to M759, D762, P765, V768, S769, Q772, and L783 in bovine AE1), R760C and S762C (R778C and S780C in bovine AE1) mutants demonstrated a greater impairment of transport (<10% activity). Human AE1 mutations at residues R760, and S762 have been previously associated with hereditary stomatocytosis and spherocytosis[1]. The location of these residues in the TM11/IL5 scaffold of bovine AE1 is shown in Fig. 6. Our functional mutagenesis data coupled with the structural and computational modeling results strongly suggest that this area is not directly involved in ion coordination and permeation but likely plays a role in the OF-IF transformation.

**CryoEM resolves symmetrical and asymmetrical dimers of AE1**. Our cryoEM data demonstrated that AE1 is a homodimer confirming previous biochemical crosslinking and hydrodynamic studies[32,33]. Transporters with the same fold as AE1 have been

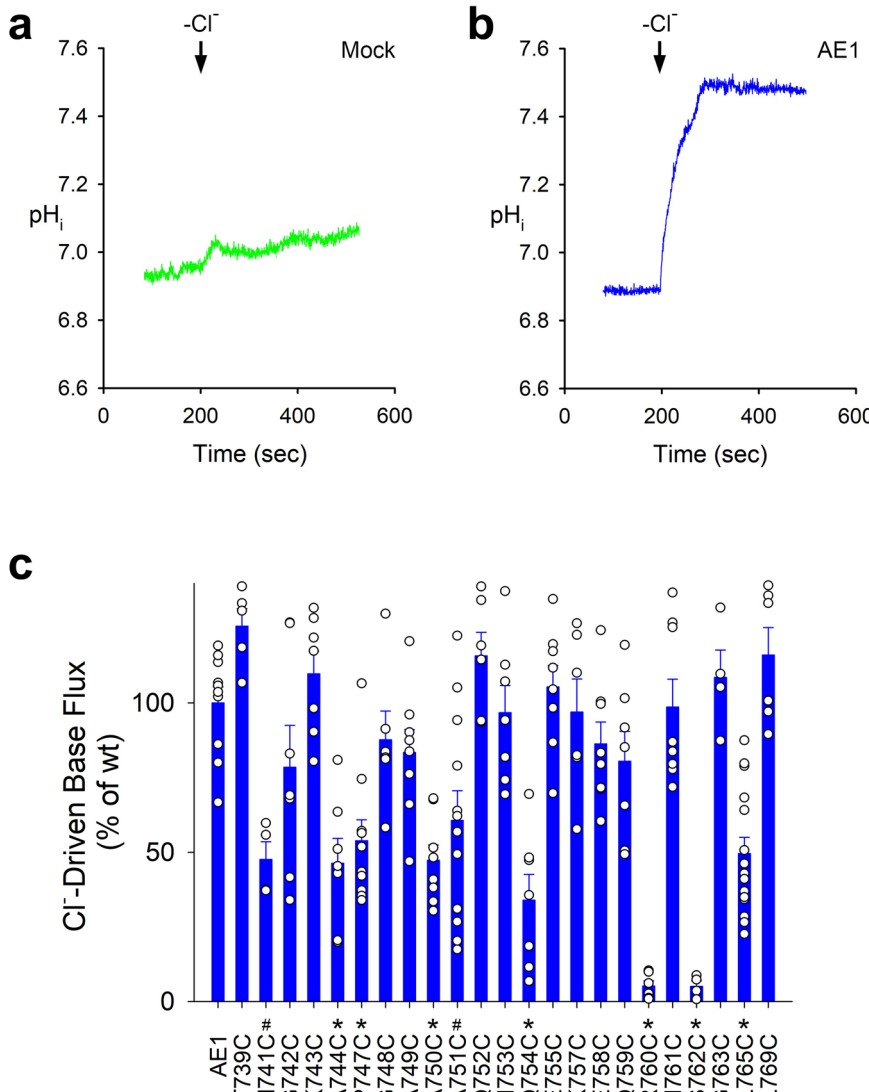

**Fig. 5 Cl⁻-driven base transport.** Typical functional traces of mock **a** and wt-AE1 **b** transfected HEK293 cells. **c** AE1 wt ($n = 10$ biologically independent experiments) and single cysteine functional mutant data (depicted as percent of wt-AE1): T739C ($n = 5$, $p = 0.4130$); M741C ($n = 4$, $p = 0.0031$); G742C ($n = 7$, $p = 0.5074$); K743C ($n = 7$, $p = 0.9817$); A744C ($n = 7$, $p < 0.0001$); P747C ($n = 10$, $p = 0.0003$); G748C ($n = 6$, $p = 0.9531$); A749C ($n = 8$, $p = 0.7563$); A750C ($n = 8$, $p < 0.0001$); A751C ($n = 12$, $p = 0.0019$); Q752C ($n = 6$, $p = 0.8550$); I753C ($n = 7$, $p = 1.0000$); Q754C ($n = 7$, $p < 0.0001$); E755C ($n = 8$, $p = 0.9993$); K757C ($n = 6$, $p = 1.0000$); E758C ($n = 8$, $p = 0.8870$); Q759C ($n = 7$, $p = 0.6288$); R760C ($n = 6$, $p < 0.0001$); I761C ($n = 8$, $p = 1.0000$); S762C ($n = 4$, $p < 0.0001$); G763C ($n = 4$, $p = 0.9968$); L765C ($n = 15$, $p < 0.0001$); and L769C ($n = 6$, $p = 0.8466$). One-way ANOVA and Dunnett's test were used to compare multiple study group means with wt-AE1. Statistically significant results differing from wt-AE1 are depicted as mean ± SEM (#$p < 0.005$ and *$p < 0.001$). Open circles represent individual data points. Source data are provided as a Source Data file.

resolved as both monomeric and dimeric (in the case of UraA, depending on the crystallization conditions)[15,34] or solely dimeric[11,13,14]. The available SLC4 structures (human AE1, human NBCe1, and rat NDCBE) have been resolved as dimers[9,11,12]. SLC26Dg is dimeric in proteoliposomes but monomeric in the presence of maltoside detergents[16]. We have previously hypothesized that the increased interaction of two gate domains in the dimeric structures might confer a greater transporter stability in the plane of the membrane during the transport cycle[11].

Our data is an additional example of the ability of cryoEM to sort particles from a heterogenous dataset[35,36]. The resolved structures demonstrate the existence of asymmetric AE1 dimers with one IF and one OF monomer. In this way, AE1 resembles the prokaryotic glutamate transporter homolog Glt_{Ph} that has three monomers, which can adopt different conformations[37].

We have previously demonstrated that there is a flexible linker between CD and TMD in the AE1 monomer that may play a significant role in the red blood cell elasticity, providing a mechanism to connect the cytoskeleton (given that AE1 CD binds various cytoplasmic proteins) with the red blood cell membrane[8]. This hypothesis is supported by the near atomic resolution data obtained in this study showing large changes in the position of the CD relative to the TMD and the recent characterization of AE1 complexes with ankyrin and protein 4.2 by cryoEM[38,39].

**Ion dynamics and implications on rate of transport.** The SILCS and MD ion and water maps (Fig. 4, Supplementary Figs. 6 and 7) demonstrate that both the IF and OF states possess large, unobstructed and heavily hydrated permeation cavities, which allow easy ion access to relevant residues from the binding pocket both from the extracellular and intracellular solutions, consistent

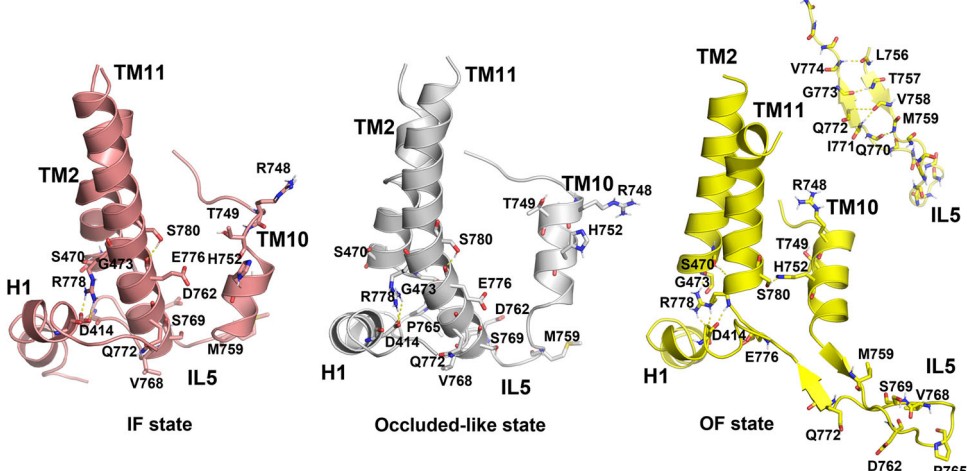

**Fig. 6 Reorganization of TM11 and IL5 during IF to OF transition in AE1.** Representative structures from MD simulations demonstrating changes in the area of TM10, IL5, and TM11 during the transition from IF to OF state: IF structure, showing the elongated TM11 (left, salmon helices); occluded-like state obtained during MD simulation after TM10 moves within the IF cavity (see Supplementary Videos 4,5) and TM11 bends at R778 (center, white helices); OF state demonstrating β-hairpin organization of IL5 (right, yellow helices and β-hairpin). The backbone atoms of IL5 forming the β-sheet are also shown as an inset. The H-bond networks between relevant residues are shown with dashes. Relevant residues studied with functional mutagenesis in this work or implicated in pathological states, are shown as sticks.

with alternating access transport[40]. Ions diffuse to the protein center aided by the presence of a number of charged residues lining the cavities (Fig. 4). In the OF state, the anions permeate downward to the area of R748, while in the IF state, where the R748 residue has been exposed to the intracellular solutions, ions move laterally towards R748 from reservoirs located at the dimeric interface. Residue E699, corresponding to residue E681 in human AE1, which has been implicated as a potential protonation site during human AE1 transport[41], is also accessible both from the extracellular and intracellular side (Fig. 4, Supplementary Fig. 6) and can potentially respond to pH changes on both sides of the plasma membrane. While the difference in the ion dynamics in both conformational states is not currently fully understood, the predicted ion reservoirs at the intracellular side might increase the probability of anion permeation into the IF cavity before the formation of a bound and occluded structure and the IF to OF transitions take place.

**The OF to IF transition in AE1 follows an elevator-like mechanism.** Comparison of the OF and IF conformations reveals significant changes in the position and shape of the core region (Figs. 3 and 6, Supplementary Figs. 4 and 9) indicating that AE1 utilizes a vertical motion (Supplementary Movie 1) accompanied by rotation of the core with respect to the gate. The elevator-like motion is also observed comparing the AE1 IF structure with the Bor1 occluded structure[13] (Supplementary Fig. 9) and is in agreement with the recent hypothesis by Ficici and coauthors that used the IF structure of AE1 generated by repeat-swap homology modeling[10]. The OF cavity and the access to the β-strands at the N-termini of TM3 and TM10 (where the putative binding sites are located), are blocked by TM3 in the IF bovine AE1 and occluded Bor1 structures and are opened only when TM3 moves upwards and away with respect to the gate (as seen in the OF bovine AE1 structure). The IF cavity and the access to the N-termini of TM3 and TM10 is blocked by TM10 in the OF bovine AE1 and (to some extent) in the occluded Bor1 state. TM10 must move downwards and away with respect to the gate (as seen in the IF bovine AE1 structure) for the IF cavity to open. The elevator-like motion of the core allows the protein center, where key binding residues such as R748 and

E699 (R740 and E681 respectively in human AE1) are located, to be consecutively exposed to the extracellular and intracellular solutions (Fig. 3, Supplementary Figs. 6 and 9), as expected for alternating transport. All in all, these three structures seem to show sensible progression from OF through occluded to IF state. It has previously been suggested that Bor1[13] and UapA[14] also utilize an elevator transport mechanism exemplified by Glt$_{Ph}$[42]. However, by comparing UraA in the IF and occluded conformation with the IF UapA structure, it was predicted that NAT family members undergo a combination of both rocking bundle and elevator transitions[34]. Rocking bundle transport motion has been suggested also for Bor1p transporter from *Saccharomyces mikatae* solved to ~6 Å on the basis of computational modeling[43]. Although the rotation of the core with respect to the gate bears some resemblance to a rocking bundle mechanism, the rigid vertical shift of the core and ion coordinating residues within the lipid membrane plane, the clear and distinct separation of the core and gate domains, and the small reorganization in the gate domain (RMSD ~0.99 Å between the gates in IF and OF state), which does not indicate complex gating rearrangement in this domain, are typically observed in elevator transporters[44]. The vertical shift of the core ion coordinating residues during the AE1 OF-IF transition is ~ 5 Å, which is similar to the coordination site shift observed in some elevator transporters[44]. The more modest protein reorganization during the elevator-like motion of the core might be advantageous in contributing to the observed high rate of transport by AE1 (turnover rate ~ 50,000/s).

**TM11 and IL5 shift between meta-stable α-helical and β-hairpin states during the OF to IF transition in AE1.** When AE1 transforms from the OF to the IF state, the length of TM11 drastically increases by incorporating residues that are part of IL5 in the OF state. This is accompanied by dissolution of the β-hairpin structure of IL5 (Figs. 6 and 7, Supplementary Fig. 4). This elongation of a TM in a native protein has not been previously observed in any of the available IF structures of 7 + 7-fold transporters. Changes in the length or conformation of α-helices have previously been reported in the context of heparin induced activation of antithrombin, which is associated with a two-turn elongation of

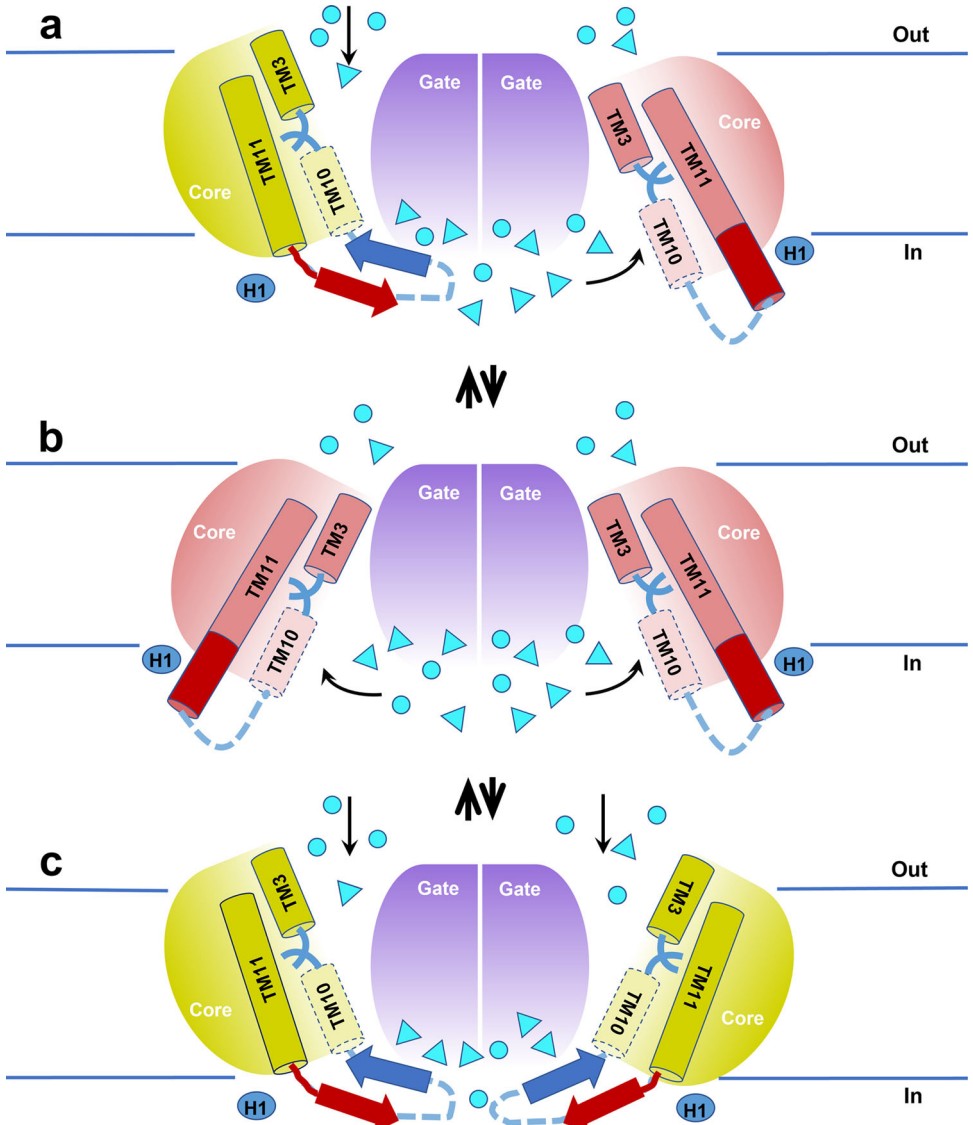

**Fig. 7 Schematic representation of the transport mechanism of bAE1.** Coexisting mixed OF–IF **a**, IF–IF **b**, and OF–OF**c** dimers. The relevant TMs (3, 10, and 11) are shown as cylinders. The core domain shifts up and down and rotates in an elevator-like motion with respect to the gate, which uncovers the center of the protein at the cross points of TMs 3 and 10 to either the extracellular or the intracellular solution. The elevator-like motion is combined with α-helical to β-hairpin conformational changes in TM11 and IL5 between TMs 10 and 11. The area of TM11 which unfolds and forms part of the β-hairpin (thick blue and red arrows) of IL5 in the OF state is shown as a red cylinder. The position of H1 with respect to TM11 in the IF and OF states is also included in the figure. The $HCO_3^-$ and $Cl^-$ ions are shown as cyan triangles and spheres, respectively, and their entry pathways in the protein center in the IF and OF monomers are indicated with black arrows. In the IF state the anions accumulated in the reservoir at the dimeric interface at the intracellular side of the protein move laterally into the IF cavity. In the OF monomers the anions diffuse vertically into the OF cavity directly from the extracellular solution.

helix D in its C-terminus[45], activation of the bacterial thermosensor DesK that involves elongation of its transmembrane helix[46], and T cell receptor major histocompatibility complexes (MHC) interaction that involves MHC alpha helices conformational changes[47]. Folding/unfolding dynamics of α-helixes plays an important role in conformational transitions of proteins. More specifically, the α-helix to β-hairpin transformation has been observed and characterized in various peptides related to neurodegenerative diseases, such as Alzheimer's, Parkinson's and Huntington's disease as well as prion diseases like Creutzfeldt-Jacob's disease and other spongiform encephalopathies[48,49]. Experimental and computational studies of model polypeptides, such as polyalanine, polylysine, and poly-glutamate have demonstrated the thermodynamic underpinnings of this transition and its dependence on environmental factors, such as temperature, pH, and salt concentrations[50,51]. In the context of the

SLC4 family, a switch between two meta-stable states (α-helix and β-hairpin) of the IL5 might facilitate and accelerate the observed OF to IF transition, with unusual TM11 elongation. In the available structures of SLC4 proteins in the OF state (human AE1, human NBCe1, rat NDCBE)[9,11,12], IL5 is fully resolved only in NDCBE. In all structures, although antiparallel β-hairpin conformation is not explicitly assigned due to the somewhat disordered organization of this region (Supplementary Fig. 4), the two antiparallel strands of IL5 are positioned close enough to provide the opportunity for formation of the H-bond network characteristic for a β-hairpin configuration and are assigned as such by PSIPRED analysis[52] (Supplementary Fig. 10). In our MD simulations of bovine AE1, IL5 naturally forms an antiparallel β-hairpin structure (Fig. 6) in all studied OF monomers when it is allowed to equilibrate and relax in the intracellular solution.

Several pathological mutations in human AE1 resulting in hereditary spherocytosis (R760Q/W, corresponding to bovine AE1 R778Q/W) or stomatocytosis (R730C, S731P, H734R, E758K, R760Q, S762R, corresponding to bovine AE1 R748C, T749P, H752R, E776K, R778Q, S780R, respectively)[1,2] are found in TMs 10 and 11 clustered in the area that reorganizes during the OF to IF transition and the opening of the IF cavity of bovine AE1 (Fig. 6). Mutation of residues in this area (Fig. 5) leads to significant decrease of transport activity, providing additional support for the potential role of TM11 and IL5 in the structural reorganizations during the OF to IF transition. The human AE1 Del400-408 (corresponding to Del418-426 in bovine AE1) mutation related to Southeast Asian ovalocytosis[1,2] deletes residues on H1 and TM1, which are also in the vicinity of the transport relevant cluster of residues of TMs 10, 11, IL5, and H1 (Fig. 6). In addition, several mutations related to hereditary stomatocytosis (R730C, S731P, H734R), characterized by increased cation leak[1,2], involve residues on TM10 from the central binding site S1 or in the vicinity of site S1 of TM10, where mutations are expected to have direct impact on ion binding[1,9,27]. Two of the human AE1 residues whose mutation to cysteine severely impede transport (R760 and S762, Figs. 5 and 6, which correspond to R778 and S780 in bovine AE1) are positioned, at the critical junction of TM11 just before the area that folds/ unfolds during the IF to OF transition. Residue S780 is involved in a H-bond with the transport essential H752 (H734 in human AE1) from TM10 in the OF state of AE1 (Fig. 6). In the IF state, TM10 is shifted down and this H-bond is broken. Instead, the OH group of S780 forms a H-bond with the backbone carbonyl oxygen of E776 (E758 in human AE1), which might stabilize the area of TM11 before the junction where it unfolds in the OF state. The observed interaction between S780 and H752 opens the possibility for potential local protonation/deprotonation effects modulating the H-bond between these two residues. R778 forms a long-lived network of H-bonds with the backbone carbonyl oxygens of S470 and G473 (S452 and G455 in human AE1) of TM2 and the carboxylate group of D414 (D396 in human AE1) of H1 in the OF state, which keep the long arginine side chain tucked underneath TM2 (Fig. 6). In the IF state, the carboxylate group of D414 of H1 forms a stable salt bridge with R778, as the arginine side chain moves away from TM2 (Fig. 6). TM11 unfolds right below R778 in the IF to OF transition. Thus, R778 and the sturdy H-bond network it forms with residues S470, G473, and D414 might be required to stabilize the helical portion of TM11. H1 is further linked via a flexible linker to the cytoplasmic domains, which connect to the cytoskeleton of the erythrocytes and influence their overall shape[1,2,8]. Mutation in these areas may therefore impact the erythrocyte shape by altering the concerted motion of H1, TMs 10 and 11, and IL5 during the catalytic cycle of AE1. Although H1 was not clearly resolved in the cryoEM structure reported here, it was included in our MD simulations. In the OF state H1 is positioned under TM2 and the shortened TM11. The elongation of TM11 necessitates a shift in H1 to provide additional space for the elongated portion. Alternatively, in the IF to OF transition, sideways motion of H1 toward TM11 may induce bending of TM11 (Supplementary Movies 4,5), leading to an occluded-like state, which precedes the α-helical unfolding and the consecutive β-hairpin formation, Fig. 6. We did not observe unfolding of TM11 in the IF monomers during the 1 microsecond MD simulations. Most of the remaining residues which impact transport, albeit not as strongly as R778 and S780 (Fig. 5), are located along the β-hairpin of IL5 (in the OF state). The β-hairpin is meta stable and is held together by H-bonds formed between backbone amide and carbonyl groups. One possible configuration of the β-hairpin is shown on Fig. 6. During MD simulations, the two antiparallel strands of the β-hairpin can

shift with respect to one another and rearrange the H-bond network accordingly, providing additional flexibility for reorganization of this region. The role of the side chains of the residues involved in the formation of the β-hairpin represents an interesting avenue for future explorations.

We propose that the TM11-IL5 transformations oscillate between metastable α-helix and β-hairpin states, which allows very rapid OF ↔ IF conversions during the transport cycle (Supplementary Movies 1–3, Fig. 7). We also propose that the transporter is predominantly in the OF conformation. This is supported by the free energy landscapes generated from coarse grained metadynamics simulations (Supplementary Fig. 11), which show a global energy minimum corresponding to the OF state and an IF state at higher energy (~5 kcal/mol above the OF state), in line with the observation that all previously resolved SLC4 structures were resolved in OF conformation[9,11,12]. AE1 is bidirectional ($HCO_3^-$ import coupled with $Cl^-$ export or $Cl^-$ import coupled with $HCO_3^-$ export), depending on the existing ion gradients. Both substrate ions likely bind at the same residues at the protein center as suggested by previous SILCS and MD simulations[27]. In the OF state, the binding of the substrate ion at the ion coordination site allows the transformation of the OF to the IF state, which includes the elongation of TM11 and the rearrangement of the β-hairpin of IL5. In the IF conformation, the elongated α-helical TM11 is unstable due to the incorporation of charged and polar residues. In addition, the potential accumulation of anions at the dimeric interface allows for their swift lateral entry into the IF cavity. This likely accelerates the IF to OF transition, during which the α-helical structure of TM11 quickly unfolds and reforms into a β-hairpin with IL5, once the required substrate ion binds to the IF state (again, depending on the existing ion gradients) (Fig. 7). The results of a study of the spring-mechanical properties of alpha helical polyglutamic acid using atomic force microscopy[53], and other studies investigating the forces within molecules such as motor proteins[54] and the forces involved in ligand receptor binding[55] and unfolding of proteins[56] support this hypothesis. Whether any of these mechanisms are used by AE1 is an attractive goal of future studies given that the turnover rate of AE1 is very close to the transport rates of channels[1,2,19]. It is currently not known whether other SLC4 transporters mediate their transport at rates similar to AE1. From this standpoint, it is interesting to note that mutation of the residues near TM11 in NBCe1 OF[57] have very similar inhibiting effect as the corresponding mutations in AE1 determined in this study, suggesting that NBCe1 may use at least in part a similar α-helix to β-hairpin switch mechanism.

## Conclusions

In this work we present the cryoEM structures of full-length bAE1 and the capture of its TMD in both IF and OF state. Notably, dimers with subunit TMDs in mixed IF–OF conformation were also detected, implying that the two monomers in the dimer can move independently from one another. The structures indicate that bovine AE1 follows an elevator-like mechanism during its transport cycle, which features slight rotation and vertical shift of the protein core with an overall core displacement of less than 5 Å and an overall vertical shift of the ion coordination site of ~5 Å. Reorganization of IL5–TM11 leads to metastable states in which IL5 is either organized as a β-hairpin (in the OF state) or folds into an α-helix leading to elongation of TM11 (in the IF state). The transported anions permeate vertically from the extracellular solution into the OF cavity in the OF state and accumulate at the dimeric interface at the intracellular side from where they can access laterally the IF cavity in the IF state. The modest protein reorganization during transport, the

oscillation between two meta-stable states of IL5–TM11 and the easy anion access into the IF and OF cavities of AE1 may explain the observed high transport rates of this protein.

## Methods

**Isolation of AE1 from bovine blood.** Bovine AE1 was purified from red blood cell ghosts isolated from defibrinated bovine blood (Quad Five) as we previously reported[8]. AE1 was extracted from the ghosts with dodecyl maltoside and purified using our method[8] except a 5 ml HiTrap ANX FF column (GE Healthcare) was used instead of a 3.5 cm DE-52 DEAE column (Whatman).

For preparation of samples for cryoEM, AE1 was mixed with amphipol PMAL-C8 (Affymetrix) at 1:3 (w/w) dilution with gentle agitation overnight at 4 °C. Detergent was removed with Bio-Beads SM-2 (Bio-Rad) incubated with samples for 1 h at 4 °C, and the beads were subsequently removed by centrifugation at 2000 g for 5 min. Amphipol containing protein was further purified on a Superose 6 column in 20 mM Tris-HCl, pH 7.5, 150 mM NaCl. The peak corresponding to dimeric AE1 was used for cryoEM analysis.

**$Cl^-/HCO_3^-$ transport assays.** HEK293 cells were transfected with wt-human AE1 cloned into a pcDNA3.1(+) expression vector, the empty vector, and specific human AE1 TM11 mutant constructs. Twenty-four hours later, the transport assays were performed. Twenty-four hour following transient transfection, pHi was measured with the various constructs in HEK293 cells grown on PEI coated coverslips as described[11,12,27]. The cells were bathed initially in the absence of $Na^+$ in a $Cl^-$-containing bicarbonate-containing solution: 115 mM tetramethylammonium chloride, 2.5 mM $K_2HPO_4$, 1 mM $CaCl_2$, 1 mM $MgCl_2$, 24 mM tetra-methylammonium bicarbonate, 5% $CO_2$, pH 7.4, and 30 μM 5-(N-ethyl-N-iso-propyl)-amiloride (EIPA). After a steady state, AE1 mediated transport was induced by switching to the following $Cl^-$-free solution: 115 mM tetra-methylammonium hydroxide, 115 mM gluconic acid lactone, 2.5 mM $K_2HPO_4$, 7.5 mM calcium gluconate, 1 mM magnesium gluconate, 24 mM tetra-methylammonium bicarbonate, 5% $CO_2$, pH 7.4, and 30 μM 5-(N-ethyl-N-iso-propyl)-amiloride (EIPA). The total cell buffer capacity (intrinsic ($\beta_i$) plus bicarbonate ($\beta_{HCO_3}$)) was calculated as described[11,27]. The rate of change of $pH_i$ ($dpH_i \, dt^{-1}$) was measured in the initial 10–15 s after a bath solution switch and converted to the rate of change of $[H^+_{in}]$ ($d[H^+_{in}] \, dt^{-1}$). The $H^+$ flux (mM $s^{-1}$) for each construct was calculated as ($\beta i + \beta HCO_3$) × ($d[H^+_{in}] \cdot dt^{-1}$).

**Sulfo-NHS-SS-biotin plasma membrane labeling.** Plasma membrane proteins were labeled and pulled down with Sulfo-NHS-SS-biotin. In this protocol the cells were washed with PBS (room temperature, pH 8.0) 24 h following transfection with various constructs. The cells were incubated (4 °C for 30 min, pH 8.0) with 1.1 mM sulfo-NHS-SS-biotin (Thermo Fisher Scientific). The reaction was then stopped using 50 mM Tris buffer at 4 °C (140 mM NaCl, pH 8.0). The cells were collected and washed with PBS, and lysed on ice in 150 mM NaCl, 0.5% sodium deoxycholate (Thermo Fisher Scientific), 1% (vol/vol) Igepal (Sigma-Aldrich), 10 mM Tris·HCl, 5 mM EDTA (Sigma-Aldrich), pH 7.5, with protease inhibitors (Roche Life Sciences). The insoluble material was pelleted over 10-min (centrifugation at 20,000 g, 4 °C). The supernatant containing >90% of the plasma membrane protein fraction was collected and incubated on a rotating shaker (4 °C for 4 h) with streptavidin-agarose resin (50 μl) (Thermo Fisher Scientific). To elute the bound proteins, the resin was pelleted and washed with the lysis buffer (60 °C for 5 min) with 2 × SDS buffer containing 2% 2-mercaptoethanol (EMD Millipore, Billerica, MA).

For the lysate detection, the cells were lysed in lysis buffer containing 150 mM NaCl, 0.5% sodium deoxycholate (Thermo Fisher Scientific), 1% (vol/vol) Igepal (Sigma-Aldrich), 10 mM Tris·HCl, 5 mM EDTA (Sigma-Aldrich), pH 7.5). The human AE1 constructs were pulled down using the AE1 2-M anti-human monoclonal antibody (Alpha Diagnostics) (1:1,000 dilution).

**SDS-PAGE and immunoblotting.** The sample proteins were resolved using 7.5% polyacrylamide gels and then transferred to polyvinylidene difluoride membranes. The expression levels of the pulled-down biotinylated proteins and whole cell lysates were determined by probing the blots with the mouse monoclonal AE1 2-M antibody (1:10,000 dilution) in TBSTM buffer (0.1% (vol/vol) Tween 20; 137 mM NaCl, 20 mM Tris, pH 7.5), containing 5% (wt/vol) nonfat milk). After 1 h incubation (room temperature), the blots were washed with TBST and then probed with Peroxidase AffiniPure Donkey Anti-Mouse IgG (H + L) (Jackson ImmunoResearch Laboratories, Inc.) at 1:10,000 dilution in TBSTM buffer and incubated at room temperature for 1 h. The blots were washed with TBST and signals were detected with ECL Western Blotting Detection Reagent (GE HealthCare).

**Electron microscopy sample preparation and imaging.** For electron microscopy of negatively stained protein, 2 μl of bovine AE1 (~0.1 mg ml⁻¹) was applied to a glow-discharged EM grid covered with a thin layer of carbon film. After 10-s incubation, the grid was stained with 0.8% uranyl formate. For cryoEM, 3 μl of bovine AE1 (~0.4 mg ml⁻¹) was applied to a glow-discharged Quantifoil 300-mesh

R1.2/1.3 grid. The grid was blotted with filter paper to remove excess sample and flash-frozen in liquid ethane with FEI Vitrobot Mark IV.

Multiple cryo electron microscopes have been used during the course of this project. At the early state, negative stain and cryoEM micrographs were recorded on a TIETZ F415MP 16-megapixel CCD camera at 50,000× nominal magnification in a FEI Tecnai F20 electron microscope operated at 200 kV. Micrographs were saved by 2x binning to yield a calibrated pixel size of 4.41 Å. Subsequently, we used a Titan Krios instrument with Gatan K2 direct electron detection camera to record cryoEM data as movies.

The frozen-hydrated grids were loaded into a FEI Titan Krios electron microscope operated at 300 kV for automated image acquisition using Leginon[58]. Micrographs (dose-fractionated movies) were acquired with a Gatan K2 Summit direct electron detection camera operated in the super-resolution mode (7,676·7,420 pixels) at a calibrated magnification of 36,764 and defocus values ranging from −1.4 to −3.2 μm. A GIF Quantum LS Imaging Filter (Gatan) was installed between the electron microscope and the K2 camera with the energy filter (slit) set to 20 eV. The dose rate on the camera was set to ~8 e⁻ pixel⁻¹ s⁻¹ and the total exposure time was 12 s fractionated into 48 frames of images with 0.25 s exposure time for each frame. Total of 3378 micrographs were collected.

**Image processing.** The frame images of each micrograph were aligned and averaged for correction of beam-induced drift using MotionCor2[21]. The local motion within a micrograph was corrected using 5 × 5 patches. Two average images, with and without dose-weighting, from all except the first frame were generated with 2× binning (final pixel size of 1.36 Å on the sample level) for further data processing. A total number of 6,392 good micrographs were picked for image processing by visual inspection of the average images and power spectra after the drift correction.

The defocus values of the micrographs were measured on the dose-unweighted average images by CTFFIND4[59]. The dose-weighted average images were used for particle picking and subsequent image processing. A total of 2,635,578 particles were automatically picked using Gautomatch[60] and windowed out in 192 × 192 pixels. Boxed particles were first subjected to 3D classifications by GPU-accelerated RELION-2[61,62], using an oval-shaped disk low-pass filtered to 60 Å as the initial model. The particles were separated into 5 classes for 34 iterations with C2 symmetry applied and the best class contained 1,115,672 particles, which were then sent to a second run of 3D classification to be sorted into 5 classes with a soft mask of the TM region applied. A number of 592,176 particles were found in the best classes after 166 iterations of the second run of 3D classification. The particles were selected and sent to 2D classification using RELION and sorted into 100 classes for 30 iterations. A total number of 405,042 particles that show good features were selected and sent to 3D classification without mask and symmetry applied. Three classes showed well resolved transmembrane helices with or without cytoplasmic domain (CD). Class 2, class 3 and class 5 with different states of CD were refined separately at first.

Class 5 showed one monomer in IF state and the other in OF state. 87,780 particles were selected and refined to a resolution of 6.9 Å. Class 3 showed both monomers in IF state. 90,818 particles were selected and refined to 6.3 Å. Class 2 without CD and class 3 with CD featured both monomers in IF states. 251,871 particles were selected and combined from class 2 and 3. The 251,871 particles were refined using a soft mask of TM region and the resolution was estimated to be 4.4 Å by RELION-2 using the "gold-standard" FSC at 0.143 criterion. The final cryoEM map was sharpened with B-factor and low-pass filtered to the stated resolution using RELION-2. The local resolution was calculated by ResMap[63] using two cryoEM maps to independently refine from halves of the data.

**Model building.** The overall IF 4.4 Å cryoEM map was inspected in Coot[64]. Aromatic residues were clearly visible in transmembrane helices (TMs) 1–9, TMs 11–14 and were used as landmarks for the following model building. The crystal structure of human AE1[9] (PDB code: 4YZF) was used as initial model. First, the crystal structure was rigid fitted into the 4.4 Å cryoEM map in UCSF Chimera[65]. The molecular dynamics flexible fitting (MDFF)[66] method was used to flexibly fit the atomic structure into the cryoEM density map in VMD. The fitted structure was inspected in Coot and residue assignment was corrected based on landmarks of aromatic residues. The loops between TMs were similarly refined in Coot without the α-helix restraints. The loop between TM8 and TM9 (residues 690 to 701), TM10 together with the loop connecting TM10 and TM11 (residues 730 to 756) were not built because of their flexibility. The atomic model of AE1 TMD was subjected to further global refinement with simulated annealing using the real space refinement feature (RSR) in the PHENIX software package[67]. Secondary structure and geometry restraints were applied to prevent structure over-fitting. Residues with poor rotamer or considered a Ramachandran outlier were fixed in Coot. Another round of RSR was performed and MolProbity analysis[68] was used to validate the final model. The OF structure was built similar to the approach described above. The CD was not built because of insufficient resolution. The same approach was used to build the TMD of the full-length IF-IF structure. The CD of full-length IF-IF was built by flexibly fitting and editing the crystal structure of the cytoplasmic domain of human AE1[25] (PDB code 4KY9). The model was further refined using RSR and validated by MolProbity analysis.

**SILCS simulations**. The Site Identification by ligand Competitive Saturation (SILCS) method[31] makes use of a combination of Monte Carlo and MD simulations in order to "flood" a protein with high concentration of small organic fragments for identification of areas within the protein matrix where the fragments tend to bind, assessed by grid free energy (GFE) maps. SILCS has been used previously for identification of ion pathways and binding sites in SLC4 proteins such as human AE1, human NBCe1, and rat NDCBE[12,27]. Detailed protocol of the SILCS simulations can be found in Ref. [69]. SILCS Software (Site Identification by Ligand Saturation, version 2020.2)[31] was utilized in order to construct the grid 3D-free energy based maps (GFE maps) that describe plausible high affinity regions of the bovine AE1 transporter which attract chemically different ions/molecules.

SILCS simulations consist of a series of calculations initiated with embedding the protein in a 120:120 Å POPC membrane bilayer with a 9:1 lipid:cholesterol mixture, followed by solvation of the systems with TIP3 water (molarity of ~55 M) and eight different fragments (benzene, propane, methanol, formamide, acetaldehyde, imidazole, methylammonium, and acetate; molarity of ~0.25 M per fragment), randomly. 10 discrete systems were prepared as described and a series of simulations of each of them was started in conjunction with the GROMACS (v. 2018) simulation package[70]. First, systems were energy minimized and equilibrated with a six-step scheme where constraints imposed on protein backbone, sidechain and lipid heavy atoms were gradually relaxed. Then, 5 ns MD simulations were run with 50.208 $kJ\cdot mol^{-1}\cdot nm^{-2}$ harmonic force constants on the protein $C\alpha$ atoms followed by 25 steps of Grand Canonical Monte Carlo (GCMC) simulations where water and probe molecules were inserted, deleted, translated or rotated following Metropolis criteria[69]. Afterwards, 100 steps of hybrid GCMC/MD simulations were done with an MD step between the GCMC steps to enhance the conformational sampling. Considering the average of the simulation data generated from the 10 discrete simulations, certain solute atoms were binned onto a grid of 1 $Å^3$ voxel followed by a Boltzmann-based transformation to yield the GFE maps representing the affinity sites of the proteins targeted by the probe solutes[71]. The convergence of the simulations was evaluated by an overlap coefficient value which was found to be 0.70 in all cases, indicative of appropriate sampling.

Throughout the simulations, CHARMM36[72] and CGenFF[73] force fields, CGenFF Program[74] and LINCS algorithm were used with a 2-fs time step. Verlet cut-off scheme and Particle Mesh Ewald techniques were used to handle the Lennard-Jones and Coloumbic interactions. The MD protocol of GCMC/MD consists of 5,000 steps of energy minimization with steepest descent followed by a 100 ps equilibration and a 1 ns production run in NPT ensemble coupled with Nose-Hoover and semi-isotropic Parrinello-Rahman temperature and pressure controlling schemes, respectively. A total simulation time of 1 μs (10.1 ns·100 steps) was achieved at 300 K and 1 bar.

**MDFF refinement of the dimers**. Prior to the MD simulations, MDFF method[66,75] was applied for the refinement of the cryoEM structures of bovine AE1. A preliminary bovine AE1 structure with included H1 helix was used for the modeling studies. The position of the H1 helix in bovine AE1 was modeled after its position in human AE1[9] and further optimized using the cryoEM maps as described below. To couple the structures with their corresponding cryoEM density maps, MDFF Module in CHARMM-GUI[75] was used for the generation of three-step protocol input files where coupling strength factor values were gradually increased as 0.3, 0.5, and 0.7. Each step was run for 1 ns. Secondary structures, cis-peptide bonds and chirality restraints were used in the fitting simulations. Simulations were done with NAMD 2.13[76] at 303.15 K, controlled with Langevin dynamics, and the CHARMM36m force field[77] and 1 fs time step. 12 Å and 10 Å distances were applied for the cut-off and switching for the non-bonded interactions. Refined structures obtained from the MDFF simulations were used for the preparation of the membrane bilayer systems and further MD simulations, herein.

**Molecular dynamics simulations**. For the molecular dynamics simulations, the apo-IF-IF and apo-IF-OF dimers were embedded in a POPC bilayer, in a rectangular box solvated with an equimolar 0.75 M NaCl + 0.75 M NaHCO₃ mixture with 20 Å water layers on both sides of the membrane. Additional details for the simulation systems are provided in Supplementary Table S1. The systems were built with the CHARMM-GUI Membrane Builder and underwent a 6-step equilibration, during which position constraints on atoms in the systems were gradually removed[78]. 20 ns long unconstraint production runs in semi-iso-thermal-isobaric (NPaT) conditions, at 310.15 K and 1 atm, were then performed with NAMD 2.13[76] using the CHARMM36 force field (CHARMM36m for proteins, CHARMM36 for lipids, TIP3P for water) and the available CGenFF parameters for the $HCO_3^-$ ions[73,77]. Long-range electrostatic interactions were evaluated with the Particle Mesh Ewald method with cutoffs of 12 and 10 Å for the electrostatic and non-bonded interactions, respectively. Three replicas per dimer, with three different relaxed structures extracted from the 20 ns MD runs were prepared and submitted for 1 μs long all-atom MD simulations with the Anton2 supercomputer and the Anton2 software version 1.31.0 from D.E. Shaw Research in an NPT ensemble at 1 bar and 303.15 K (controlled with an MTK barostat coupled to the Nosé-Hoover thermostat)[27]. The nonbonded interactions were computed with the RESPA multiple-time-step algorithm. Temperature and semi-isotropic pressure

coupling were performed with the multi-integrator (multigrator) algorithm and the long-range electrostatics interactions were evaluated with the available *u*-series algorithm. Wrapping centering alignment and analysis of the produced trajectories was done with VMD 1.9.3[79]. The first 50 ns of the 1 μs trajectories were removed from the analysis. The VolMap tool of VMD 1.9.3 was used for generation of the density maps. The reported density maps were averaged over the three replicas simulated for the IF-IF and IF-OF dimers. RMSD plots calculated for the backbone protein atoms are shown in Supplementary Fig. 12 and areas corresponding to relevant conformational changes (bending of TM11 or occlusion of the IF cavity by movement of TM10) are noted with an asterisk.

**Statistics and reproducibility**. All statistical tests used, sample sizes, and the number of replicates are described in the corresponding methods, figure legends and tables.

**Reporting summary**. Further information on research design is available in the Nature Research Reporting Summary linked to this article.

## Data availability

The final cryoEM density maps of bovine AE1 IF-IF TMD, IF-IF full-length protein and IF-OF full-length protein have been deposited to the Electron Microscopy DataBank (EMDB) under the accession codes EMDB-27267, EMDB-28055, and EMDB-27856, respectively. The final atomic models of bovine AE1 IF-IF TMD, IF-IF full-length protein and IF-OF full-length protein have been deposited into the Protein Data Bank (PDB) under the accession codes 8D9N, 8EEQ and 8E34, respectively. Source data are provided with this paper. Initial and final steps from the MD trajectories and sample input files for Anton2 MD simulations are provided in the Supplementary Data 1 file. Source data are provided with this paper in Supplementary Data 2.

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

## Acknowledgements

I.K. is supported by the NIH grant R01DK077162, the Factor Family Foundation Chair in Nephrology, Smidt Family Foundation, Paula Block Charitable Foundation, and the Kleeman Foundation. Z.H.Z. was supported by the NIH grant R01GM071940. We acknowledge the use of resources at the Electron Imaging Center for NanoMachines of the California NanoSystems Institute (CNS) at University of California, Los Angeles, which are supported by grants from the NIH (1S10RR23057 and 1U24GM116792) and NSF (DMR-1548924 and DBI-1338135). We acknowledge the Department of Medicine for support of N.A. and A.P. Work in the SYN lab was supported by the NIH (R01 DK077162) and partially by the Natural Sciences and Engineering Research Council of Canada (NSERC grant No. RGPIN-2021-02439). D.P.T. acknowledges support by the Canadian Institutes of Health Research (CIHR, grants PJT-156236 and PJT-180245) and the Canada Research Chairs program. MD simulations were performed on the Canada Foundation for Innovation supported GladOS cluster at the University of Calgary and on the West-Grid/Compute Canada clusters under Research Allocation Award to SYN. Part of the production simulations were performed on the Anton 2 computer provided by the Pittsburgh Supercomputing Center (PSC) and DE Shaw Research through Grant R01GM116961 from the NIH.

## Author contributions

I.K. and A.P. were responsible for the initiation and the overall management of the project. K.T. and A.P were responsible for purification and biochemical characterization of AE1. L.K., R.A, D.N., and N.A. performed functional mutagenesis experiments. I.K. and L.K. analyzed functional mutagenesis data. J.J., W.W., and Z.H.Z were responsible for collecting the cryoEM data. J.J., W.W., Z.H.Z., and H.R.Z. were responsible for analyzing the cryoEM data and for model building and interpretation. H.R.Z., S.Yu.N., and D.P.T. planned the computational experiments. H.R.Z., G.K., and H.M.K. performed the computational experiments and analyzed the computational data. A.P., H.R.Z., W.W., Z.H.Z., D.P.T. and I.K. wrote the manuscript. All authors contributed editorial feedback leading to the final version of the manuscript.

## Competing interests

The authors declare no competing interests.
