## [Peer Review File · Communications Biology]

Reviewers' comments:

Reviewer #1 (Remarks to the Author):

Review of COMMSBIO-22-2034

This collaborative study combines cryo-electron microscopy and molecular dynamics simulations to determine the structure of the bovine anion exchanger 1 (band 3) in various conformations, including, for the first time, the inward-facing state and importantly, a dimer consisting of both the outward-facing and inward-facing state. Band 3 consists of 14 transmembrane (TM) segments with a 7+ 7 inverted topology arranged into a gate domain at the dimer interface and a core domain containing the anion binding site located between two short TM segments 3 and 10. The inward-facing state revealed an elongated TM 11 with IL5 with a 20° rotation and modest 5Å downward movement of the core domain. The structural and modelling studies were completed by functional mutagenesis that confirmed the essential role of certain residues, including hArg730. This study provides an important advance over the currently-available structures of outward facing state and supported an “elevator” mode of transport. This study also places mutations linked to disease in a proper structural context.

Comments

1. (p. 5, line 127) A 1981 paper by Macara and Cantley in Biochemistry (<https://pubmed.ncbi.nlm.nih.gov/7295667/>) provided early evidence using inhibitors that “the subunits of a band 3 dimer transport anions independently”.
2. (p. 7, Line 220) The notation of residues is confusing and should indicate the mutation of residues in bovine AE1 (and the corresponding residues in human AE1 listed in brackets).
3. (p. 8, Line 252) Reference should be made here to two recently published papers that describe the Band 3-ankyrin complex (<https://www.nature.com/articles/s41594-022-00792-w>; <https://www.nature.com/articles/s41594-022-00779-7>). In addition, point out that the cytoplasmic domain is stabilized by interaction with ankyrin and protein 4.2.
4. Discuss: <https://www.biorxiv.org/content/10.1101/2022.02.11.480130v1.full> for structures of full-length Band 3 in apo form and with inhibitors.
5. (Methods) Is the structure solved without inhibitors like H2DIDS, different from the previous methods?
6. Fig. 3c and Fig. 4a appear to be repeats.

Reviewer #2 (Remarks to the Author):

Anion exchanger 1 is an ion (HCO₃⁻/Cl⁻) exchange transporter essential for the efficient removal of carbon dioxide from the tissues. It exhibits particularly fast exchange and mutations are known to cause anemia. All structures until now were of the outward-facing state and show a 7 + 7 inverted repeat fold. Zhekova et al present structures at medium resolution (4.4 Å is not what I would call “near atomic”) of the bovine AE1 in both IF and OF states derived from single-particle cryo-EM. Furthermore, the authors also describe a heterodimer of both IF and OF states. The IF state exhibits an unusual extension of TM 11 partially formed from residues in intracellular loop 5.

The work is timely and potentially extends our knowledge of how transporters work. Technically, the work is fairly standard and I am reasonably content it has been performed well enough. However, the manuscript has significant flaws which need attention:

1. In their description of the mechanism of this transporter on pages 5 and 6, the authors seem overly keen to make it fit the elevator mechanism. Although fig 3B does shown a small downward

motion of the core domain, if one looks at fig 3C (which nicely shows the nature of the alternating access), it looks far more intuitive to call this a simple rocking bundle (as summarized by Drew et al. Chem. Rev. 2021 121:5289-5335) mechanism where one bundle is fixed. Even in the rocking-bundle model, one can find examples of where there is a small vertical shift in one sub-domain when comparing OF/IF states. Therefore, it looks to me much more like other rocking bundles and less like the other elevators where there really are large vertical shifts (like GltPh and CitS). Movie 1 is a bit misleading - the translation to move the core domain in between the two conformations is broken down into rotation and translation, but of course this will happen (as revealed by the PCA-derived movies in SI movies 2 and 3) simultaneously. The authors discuss exactly these issues in the Discussion, but to me it really feels like they are shoe-horning their structure to fit the elevator, despite acknowledging that it is really just a rocking bundle as other related studies concluded. Of course, these mechanisms can and will get blurred over time as reality will likely reflect a continuum of mechanisms and this seems like a structure is at the point of blurring to extremes and should therefore be discussed as such.

2. The second problem which adds to the confusion is that the so-called "gate" domain does not actually move. Once again intuitively, the use of the word gate implies something moves or changes at least to allow movement of ions. Of course, this is one of the problems of nomenclature in biology and its probably too late to change this, but it really does not help give an intuitive picture of what is really happening here and distracts from the message of the paper (I accept that the AE1 community have probably committed to this though).

3. Fig4a does not show meshes of SILCS maps – presumably this is supposed to SI figure 7? But even then there are no yellow meshes and later becomes apparent that figure 7 is more likely the plain vanilla MD (consistent at least with the legend). So this section is a bit of a mess and I cannot really dissect information from this at the moment. The SILCS maps are missing so we cannot interpret those, and Figs S6 and S7 don't show that much that could not have predicted just by simply looking at the structure. The authors mention that there are differences in the way the anions permeate to the different cavities in the different state, but I am not sure what the take home message is supposed to be, or whether there are testable hypotheses?

4. The discussion about the mechanism is very ambiguous in terms of ion directions and it remains unclear to me what is binding when and to what state to drive the OF \leftrightarrow IF or IF \leftrightarrow OF transitions. I understand its reversible, but do Cl⁻ and HCO₃⁻ bind to the same residues for example? This whole section also needs rewriting.

Minor points.

Line 133 – Figure 2 does not really show me the breakdown of the gate and core domains.

Line 588 " SILCS simulations consist of a series calculations initiated with embedding the protein in a 120:120 Å POPC membrane bilayer with a 9:1 lipid:cholesterol mixture, followed by solvation of the systems with TIP3 water (molarity of ~55 M) and nine different fragments (molarity of ~0.25M per fragment), randomly." What are the "nine different fragments"?

Line 595 - please remove x from the units.

Line 627 – why does the SILCS MD system have cholesterol but the vanilla MD does not?

Line 650 – I am confused by the data deposition statement – it only refers to one model? Surely there is a model for both IF-IF and OF-IF?

Line 656 – the web address is complete.

Finally, a plea - can we please have figures inline for reviewing.

Reviewer #3 (Remarks to the Author):

The manuscript of Zhekova, Jiang, and Wang describes the structure of Anion Exchanger 1 (SLC4A1)

captured in two different conformations, the outward-facing and the inward-facing. While the outward-facing structure of SLC4 proteins has been presented several times by now, it is the inward-facing structure, or specifically, the combination of revealing both structural states for one and the same protein, that is the most exciting finding novel claim of this manuscript. This combination leaves little doubt to the authors and allows them to finally confirm that the transport mechanism of SLC4 transporters is indeed the expected elevator alternating access mechanism. This claim, if convincingly demonstrated, is of significant importance for the transporter community.

The moderate resolution of the IF conformation (4.4 Å), and especially the complete lack of information on the quality of the OF structure and the CD in context of the IF structure represent severe weaknesses of this manuscript. In its current state, claims on the rotation of the CD and extension of TM11 are not justified or convincing. What remains is the comparison of the TMDs between OF human AE1 and IF bovine AE1 that confirms the elevator mechanism.

Regrettably, this implies that I do not find the manuscript acceptable for publication in its current state.

Major issues

Line 109: the authors indicate a 45 degrees rotation of the CD with respect to the TMD. However, the quality of the structure of the CD in the context of the full-length structure is not indicated, these parameters appear only to be provided for the IF TMD (Table 1). Please provide quality indicators for the CD that justify this claim or remove this statement (and other references to the full-length structures, like line 82).

Line 112: "the linker [...] is not rigid". Provided that the authors can provide convincing justification of the 45 degrees rotation (see line 109), they should indicate that the indicated flexibility of the linker was observed for detergent-solubilized protein. Whether this also holds true in the native membrane-inserted state is not clear from the current data.

Line 138: "IF and OF conformations", the major novelty of this manuscript concerns the comparison of both the IF and OF conformations for one single protein, bovine AE1. It appears that Table 1 contains quality indicators for the IF structure, though this is not indicated. Similar statistics should also be provided for the OF conformation to make this IF-OF comparison meaningful. The absence of these statistics prevents any meaningful analysis of the TM11 extension as well.

Line 217: many mutants are analyzed, but the relevance of these mutations is not made clear apart from the R760. It appears (see comment on fig. 5) that this is a sensitive region of the protein, but the link to the current manuscript is not clear. The corresponding section in the discussion (line 331) does not provide additional clarification. The authors should consider to experimentally demonstrate that these mutations affect the suspect TM11 elongation or use their computational studies to determine whether the anion clouds on the cytoplasmic side are affected.

Fig.5: the chloride flux is indicated as percentage of wild type. In the corresponding suppl. fig. 8, the surface expression levels are not indicated as percentage of wild type. This prevents a meaningful correction of the observed activity with the observed surface expression levels and thereby makes the data presented in Fig 5 not convincing.

Line 243: "decreased unidirectional rate". I have two concerns regarding this statement. 1) given that the protomers appear to move independently, it is unclear why an asymmetric dimer would lead to a reduction in transport rate. Both protomers will reorient in the presence of substrate regardless of their initial conformation. 2) "a minority of dimers" (line 245), what makes the authors believe that the observed distribution of asymmetric and symmetric dimers in amphipols is representative of the distribution present in a lipid membrane? Also keeping in mind that the structures are based on a small subset (10-20%?) of the total number of particles. Please explain or remove these claims.

Line 263: "E699", since the authors claim to have both the IF and OF conformations, they should make proper use of their MD simulations and determine whether the pKa of E699 changes significantly between both conformations in line with the suspected role as proton acceptor.

Line 268: "probability", it seems extremely unlikely that the suspected anion reservoirs increase the probability of a liganded occluded conformation. As far as I am aware AE1 is an obligatory exchanger, implying that an unliganded occluded conformation is by definition not allowed, as this would allow reorientation of the "empty" carrier. Though the mechanistic basis for this exclusion is not clear at the moment, it appears likely that this does not involve an increase in the probability of binding substrate but more in an energetic penalty that prevent the IF carrier to reach the occluded conformation. Please explain.

Fig 3C: this panel is essentially identical with the Fig 4A panel. Legend of Fig 4 does not fit with Fig 4.

Minor issues

The authors confirm the elevator mechanism for SLC4 transporters. This makes the time right to move away from the core/gate nomenclature historically established by Lu et al. 2011 and switch to the transport/scaffold nomenclature commonly used for elevator transporters as suggested by Ficici et al. 2017 (ref 37).

The overall quality of the figures is below par. E.g., fig3: the superimposition makes the structures difficult to analyze (like suppl fig3 and suppl fig4), panel b has too much detail (like fig5).

Line 84: indicate the protein was reconstituted in amphipols.

Line 131: SLC29 should be SLC23

Line 161: "exposes", please comment whether the unfolding of the beta-sheet is absolutely required for exposing R748 and E699. It appears that in the SLC26 and SLC23 IF structures changes in secondary structure are not required to expose the substrate binding site.

Line 170: this section shows large overlap with the previous section (line 147). I suggest the authors to combine these sections.

Line 182: indicate whether the TM11 unfolding is observed in the MD simulations.

Line 339 "elect", effect?

Fig 3C legend: please remove "atomic". A surface representation can never be "atomic"

Suppl fig7: please indicate what the densities mean, e.g., can this be related to a certain concentration and if so, are the contour levels identical for bicarbonate and chloride?

We thank the reviewers for the very helpful comments. We have addressed all the suggestions raised by the reviewers in the revised manuscript.

Review of COMMSBIO-22-2034

Reviewer #1 (Remarks to the Author):

Reviewer 1 mentioned that our “study provides an important advance over the currently-available structures of outward facing state and supported an “elevator” mode of transport. This study also places mutations linked to disease in a proper structural context.”

Comments

1. (p. 5, line 127) A 1981 paper by Macara and Cantley in Biochemistry (<https://pubmed.ncbi.nlm.nih.gov/7295667/>) provided early evidence using inhibitors that “the subunits of a band 3 dimer transport anions independently”.

The reference has been included in the revised manuscript as requested (Ref. 10 in the revised manuscript).

2. (p. 7, Line 220) The notation of residues is confusing and should indicate the mutation of residues in bovine AE1 (and the corresponding residues in human AE1 listed in brackets).

As stated in the methods, the functional mutations were made in human wt-AE1, and therefore the corresponding bovine AE1 residues were depicted in brackets.

3. (p. 8, Line 252) Reference should be made here to two recently published papers that describe the Band 3-ankyrin complex (<https://www.nature.com/articles/s41594-022-00792-w>; <https://www.nature.com/articles/s41594-022-00779-7>). In addition, point out that the cytoplasmic domain is stabilized by interaction with ankyrin and protein 4.2.

These changes have been made in the revised manuscript as requested (Ref. 38 and 39).

4. Discuss: <https://www.biorxiv.org/content/10.1101/2022.02.11.480130v1.full> for structures of full-length Band 3 in apo form and with inhibitors.

We have discussed these structures as requested in the revised manuscript (Ref. 18).

5. (Methods) Is the structure solved without inhibitors like H2DIDS, different from the previous methods?

The OF structure we report does not differ in any significant way.

6. Fig. 3c and Fig. 4a appear to be repeats.

Fig. 4 has been corrected.

Reviewer #2 (Remarks to the Author):

Anion exchanger 1 is an ion ($\text{HCO}_3^-/\text{Cl}^-$) exchange transporter essential for the efficient removal of carbon dioxide from the tissues. It exhibits particularly fast exchange and mutations are known to cause anemia. All structures until now were of the outward-facing state and show a 7 + 7 inverted repeat fold. Zhekova et al present structures at medium resolution (4.4 Å is not what I would call “near atomic”) of the bovine AE1 in both IF and OF states derived from single-particle cryo-EM. Furthermore, the authors also describe a heterodimer of both IF and OF states. The IF state exhibits an unusual extension of TM 11 partially formed from residues in intracellular loop 5.

The work is timely and potentially extends our knowledge of how transporters work. Technically, the work is fairly standard and I am reasonably content it has been performed well enough. However, the manuscript has significant flaws which need attention:

1. In their description of the mechanism of this transporter on pages 5 and 6, the authors seem overly keen to make it fit the elevator mechanism. Although fig 3B does show a small downward motion of the core domain, if one looks at fig 3C (which nicely shows the nature of the alternating access), it looks far more intuitive to call this a simple rocking bundle (as summarized by Drew et al. *Chem. Rev.* 2021 121:5289-5335) mechanism where one bundle is fixed. Even in the rocking-bundle model, one can find examples of where there is a small vertical shift in one sub-domain when comparing OF/IF states. Therefore, it looks to me much more like other rocking bundles and less like the other elevators where there really are large vertical shifts (like GltPh and CitS). Movie 1 is a bit misleading - the translation to move the core domain in between the two conformations is broken down into rotation and translation, but of course this will happen (as revealed by the PCA-derived movies in SI movies 2 and 3) simultaneously. The authors discuss exactly these issues in the Discussion, but to me it really feels like they are shoe-horning their structure to fit the elevator, despite acknowledging that it is really just a rocking bundle as other related studies concluded. Of course, these mechanisms can and will get blurred over time as reality will likely reflect a continuum of mechanisms and this seems like a structure is at the point of blurring to extremes and should therefore be discussed as such.

We agree with the reviewer that the movement is more complex. The AE1 OF to IF shift demonstrates a rigid core vertical motion combined with a rotation motion with respect to the gate. It is generally accepted that in rocking bundle transporters, the binding site does not shift vertically within the membrane plane in a significant way and barriers at the extracellular and intracellular sides are formed by gating reorganization in the subunits, containing the bundle. However, considering the clear spatial separation into well-defined core/gate domains in AE1, the neglectable reorganization in the gate domain (RMSD ~ 0.99 Å calculated between IF and OF gate domains), which suggests it's not involved in complex gating events, the rigid motion of the core with respect to the gate domain, and the non-neglectable overall vertical displacement of the core binding residues from site S1 (Revised Fig. 3, new Supplementary Figure 4b) within the plane of the membrane, we believe that the elevator mechanism describes better the observed conformational changes in AE1. Moreover, elevator transporters are fairly diverse and not all of them feature the classical large downward shift of the transport domain observed in GltPh. The review by Garaeva and Slotboom, 2020, Ref. 44, that we added to the revised manuscript, highlights this mechanistic diversity and summarizes additional characteristics of transporters that use an elevator mechanism. Our data supports the criteria in this review suggesting that AE1 is an elevator protein. Nevertheless, due to the more complex movements in AE1, we have replaced

the term “elevator” with “elevator-like” throughout the manuscript to reflect this ambiguity. We have also made alterations to the text to address the similarities with rocking bundle mechanism and we have replaced Movie 1 with a new movie, which better captures the vertical movement of the core domain and the binding site residues from the core versus the gate domain.

2. The second problem which adds to the confusion is that the so-called “gate” domain does not actually move. Once again intuitively, the use of the word gate implies something moves or changes at least to allow movement of ions. Of course, this is one of the problems of nomenclature in biology and its probably too late to change this, but it really does not help give an intuitive picture of what is really happening here and distracts from the message of the paper (I accept that the AE1 community have probably committed to this though).

We thank the reviewer and agree with the points raised. We agree that the gate/core nomenclature used in the literature is confusing since its original usage with regards to the UraA transporter. However, given that essentially all previous SLC4 publications have used the older nomenclature, we have decided to still refer to the original terminology to prevent confusion. In light of this and the points raised by both reviewers 2 and 3, we have added a footnote that addresses the issues with the usage of gate/core nomenclature.

3. Fig4a does not show meshes of SILCS maps – presumably this is supposed to SI figure 7? But even then there are no yellow meshes and later becomes apparent that figure 7 is more likely the plain vanilla MD (consistent at least with the legend). So this section is a bit of a mess and I cannot really dissect information from this at the moment. The SILCS maps are missing so we cannot interpret those, and Figs S6 and S7 don't show that much that could not have predicted just by simply looking at the structure. The authors mention that there are differences in the way the anions permeate to the different cavities in the different state, but I am not sure what the take home message is supposed to be, or whether there are testable hypotheses?

Fig. 4 has been corrected and now shows the mentioned SILCS and MD maps. We find the difference in anion behavior at the intra and extra-cellular side interesting, considering the inverse topology of the transporter which would imply similar behavior at both sides. We have discussed that both ion behaviors can be related to the high transport rate of the transporter, which is a hypothesis we plan to test in the future with free energy calculations of permeation barriers along the identified ion pathways.

4. The discussion about the mechanism is very ambiguous in terms of ion directions and it remains unclear to me what is binding when and to what state to drive the OF \leftrightarrow IF or IF \leftrightarrow OF transitions. I understand its reversible, but do Cl⁻ and HCO₃⁻ bind to the same residues for example? This whole section also needs rewriting.

In our previous computational work on the OF state of AE1 where we assessed in more detail the binding of Cl⁻ and HCO₃⁻, the results indicated that both ions bind at the same residues at the center of the protein (Ref. 27 in the revised manuscript). We have rewritten the text as requested to clarify the sequence of events during the IF to OF and OF to IF transitions. AE1 is bidirectional and as such can transport its substrate ions in either direction (HCO₃⁻ influx coupled to Cl⁻ efflux or Cl⁻ influx coupled with HCO₃⁻ efflux), depending on the existing ion gradients. In the OF state,

once the required substrate ion is bound at the ion coordination site, the outside-to-inside ion gradient drives this ion toward the cytoplasm and induces the transformation of the OF to IF state, which includes the elongation of TM11 and the rearrangement of the β -hairpin of IL5. In the IF conformation, the elongated α -helical TM11 is unstable due to the incorporation of charged and polar residues. In addition, the accumulation of anions at the dimeric interface allows for their swift lateral entry into the IF cavity. This likely accelerates the IF to OF transition, during which the α -helical structure of TM11 quickly unfolds and reforms into a β -hairpin with IL5, once the required substrate ion binds to the IF state (again, depending on the existing ion gradients) (Fig. 7).

Minor points.

Line 133 – Figure 2 does not really show me the breakdown of the gate and core domains.

Fig. 2 has been corrected in the revised manuscript.

Line 588 “ SILCS simulations consist of a series calculations initiated with embedding the protein in a 120:120 Å POPC membrane bilayer with a 9:1 lipid:cholesterol mixture, followed by solvation of the systems with TIP3 water (molarity of ~55 M) and nine different fragments (molarity of ~0.25M per fragment), randomly.” What are the “nine different fragments”?

We have added the requested information in the SILCS methodology section.

Line 595 - please remove x from the units.

The change has been made as requested.

Line 627 – why does the SILCS MD system have cholesterol but the vanilla MD does not?

The system setup used in the SILCS program includes cholesterol by default during system building. We did not include cholesterol in our MD simulations, since we were interested in the ion and water interactions with the proteins rather than the lipid-protein ones and we used a simplified bilayer made of POPC as a reliable medium for 1 μ s long MD simulations of SLC4 proteins which we have tested and used in previous published work. We are currently assessing lipid-protein interactions in the SLC4 family in detail with more complex membranes representing simplified HEK293 membrane models, however this analysis is outside of the scope of the current manuscript.

Line 650 – I am confused by the data deposition statement – it only refers to one model? Surely there is a model for both IF-IF and OF-IF?

We now have uploaded the full-length IF-OF and IF-IF structures in addition to the (IF-IF) TMD structure as stated in the revised manuscript.

Line 656 – the web address is complete.

It is completed as requested.

Finally, a plea - can we please have figures inline for reviewing.

The revised manuscript has been modified as recommended.

Reviewer #3 (Remarks to the Author):

The manuscript of Zhekova, Jiang, and Wang describes the structure of Anion Exchanger 1 (SLC4A1) captured in two different conformations, the outward-facing and the inward-facing. While the outward-facing structure of SLC4 proteins has been presented several times by now, it is the inward-facing structure, or specifically, the combination of revealing both structural states for one and the same protein, that is the most exciting finding novel claim of this manuscript. This combination leaves little doubt to the authors and allows them to finally confirm that the transport mechanism of SLC4 transporters is indeed the expected elevator alternating access mechanism. This claim, if convincingly demonstrated, is of significant importance for the transporter community.

The moderate resolution of the IF conformation (4.4 Å), and especially the complete lack of information on the quality of the OF structure and the CD in context of the IF structure represent severe weaknesses of this manuscript. In its current state, claims on the rotation of the CD and extension of TM11 are not justified or convincing. What remains is the comparison of the TMDs between OF human AE1 and IF bovine AE1 that confirms the elevator mechanism.

Regrettably, this implies that I do not find the manuscript acceptable for publication in its current state.

We agree that our IF map is at a moderate resolution, however our manuscript represents the first experimental data for an IF model in any mammalian SLC4 transporter coupled with extensive molecular dynamics simulations based on our data. The data we provide is accordingly an important step forward and contribution for both for the AE1 field and for the SLC4 family of transporters in general.

Major issues

Line 109: the authors indicate a 45 degrees rotation of the CD with respect to the TMD. However, the quality of the structure of the CD in the context of the full-length structure is not indicated, these parameters appear only to be provided for the IF TMD (Table 1). Please provide quality indicators for the CD that justify this claim or remove this statement (and other references to the full-length structures, like line 82).

The requested data has now been included in Table 1 in the revised manuscript.

Line 112: "the linker [...] is not rigid". Provided that the authors can provide convincing justification of the 45 degrees rotation (see line 109), they should indicate that the indicated flexibility of the linker was observed for detergent-solubilized protein. Whether this also holds true in the native membrane-inserted state is not clear from the current data.

As shown in Fig. 1, only one monomer CD domain rotates in AE1 dimer, whereas the second monomer CD domain does not, suggesting that the phenomenon is not an artifact of the presence of detergent. Furthermore, we have shown this phenomenon in the absence of detergent in our previous lower resolution EM study (Ref. 8 in the revised manuscript). Finally, the preliminary data

of Capper et al. 2022 (Ref. 18 in the revised manuscript; Supplementary Fig. 1) also supports our conclusion.

Line 138: “IF and OF conformations”, the major novelty of this manuscript concerns the comparison of both the IF and OF conformations for one single protein, bovine AE1. It appears that Table 1 contains quality indicators for the IF structure, though this is not indicated. Similar statistics should also be provided for the OF conformation to make this IF-OF comparison meaningful. The absence of these statistics prevents any meaningful analysis of the TM11 extension as well.

The reviewer is correct in that the original Table 1 reports statistics for the IF conformation. Statistics for our full-length IF-OF and IF-IF structures are now added to the revised Table 1.

Line 217: many mutants are analyzed, but the relevance of these mutations is not made clear apart from the R760. It appears (see comment on fig. 5) that this is a sensitive region of the protein, but the link to the current manuscript is not clear. The corresponding section in the discussion (line 331) does not provide additional clarification. The authors should consider to experimentally demonstrate that these mutations affect the suspect TM11 elongation or use their computational studies to determine whether the anion clouds on the cytoplasmic side are affected.

We anticipate that the main function of certain residues in this area is to mediate fast and reversible elongation of TM11 in the IF conformation, which accompanies the downward motion of the substrate coordination site residues in the TM10 area. The beta-hairpin is supported by H-bonds formed from the protein backbone with small participation of the residue side chains. In agreement with this, mutation of most of the residues in this region did not have drastic effect on AE1 function. The most affected residues, R760 and S762 (R778 and S780 in bovine AE1), were at the N-terminus of TM11 in the OF conformation. It is interesting that in NBCe1 (SLC4A4), the homologous residues were also very sensitive to mutation (Ref. 57 in the revised manuscript) suggesting that might be using the same TM11 elongation mechanism in the OF to IF transformation.

To study the detailed effect of the residues in this area on the TM11 structure structurally and/or computationally is well beyond the scope of our current manuscript, however we do plan to perform these suggested studies in the future.

Fig.5: the chloride flux is indicated as percentage of wild type. In the corresponding suppl. fig. 8, the surface expression levels are not indicated as percentage of wild type. This prevents a meaningful correction of the observed activity with the observed surface expression levels and thereby makes the data presented in Fig 5 not convincing.

As suggested by the reviewer, we now depict the surface expression data as a percent of wild-type data in Supplementary Fig. 8 in the revised manuscript. We also depict the ratio of the functional data to the surface protein expression Supplementary Fig. 8 in the revised manuscript.

Line 243: “decreased unidirectional rate”. I have two concerns regarding this statement. 1) given that the protomers appear to move independently, it is unclear why an asymmetric dimer would lead to a reduction in transport rate. Both protomers will reorient in the presence of substrate regardless of their initial conformation. 2) “a minority of dimers” (line 245), what makes the authors believe that the observed distribution of asymmetric and symmetric dimers in amphipols is representative of the

distribution present in a lipid membrane? Also keeping in mind that the structures are based on a small subset (10-20%?) of the total number of particles. Please explain or remove these claims.

We have removed the statement as suggested.

Line 263: “E699”, since the authors claim to have both the IF and OF conformations, they should make proper use of their MD simulations and determine whether the pKa of E699 changes significantly between both conformations in line with the suspected role as proton acceptor.

The reviewer raises an interesting point. Since AE1 is a bidirectional transporter, which can transport HCO_3^- bidirectionally depending on the Cl^- and HCO_3^- gradients, it is likely that protonation/deprotonation events can occur both in the IF and OF state. PropKa calculations of the pKa of E699 in the AE1 OF and IF structures yield ~ 8.0 . Given the extracellular and intracellular pH values in vivo, this predicts that E699 may be protonated in both the OF and IF conformation. PropKa is a simplified semiempirical method, which considers the geometry of the molecule, how buried the titratable residues are, and what other residues are found around them (e.g. potential salt bridges, coulomb, dipole-dipole interactions), but does not include dynamic structural or solvent effects. As such, the provided pKa values give more of a qualitative (i.e. one can make statement such as: ‘it is likely that the residue is protonated’) rather than an exact quantitative description of the protonation states. More accurate pKa calculations would require specific methods that account for structural dynamics and solvent-protein interactions, which are fairly complex and time consuming and are outside of the scope of the present manuscript. We are currently studying the protonation effects in E681 of human AE1 with a variety of computational methods, including CpHMD and QM/MM calculations, which can factor in these important structural and solvent effects and can lead to better understanding of the role of protonation during the transport cycle of SLC4. We have modified the text to reflect the bidirectionality of AE1 and the potential role of protonation/deprotonation on both sides of the membrane.

Line 268: “probability”, it seems extremely unlikely that the suspected anion reservoirs increase the probability of a liganded occluded conformation. As far as I am aware AE1 is an obligatory exchanger, implying that an unliganded occluded conformation is by definition not allowed, as this would allow reorientation of the “empty” carrier. Though the mechanistic basis for this exclusion is not clear at the moment, it appears likely that this does not involve an increase in the probability of binding substrate but more in an energetic penalty that prevent the IF carrier to reach the occluded conformation. Please explain.

We agree with the reviewer and have accordingly clarified the discussion in the revised manuscript (page 9 in the revised manuscript): While the difference in the ion dynamics in both conformational states is not currently fully understood, the ion reservoirs at the intracellular side might increase the probability of anion permeation into the IF cavity before the formation of a bound and occluded structure and the IF to OF transitions take place.

Fig 3C: this panel is essentially identical with the Fig 4A panel. Legend of Fig 4 does not fit with Fig 4.

In the revised manuscript, the correct Fig. 4 showing the SILCs maps is now included.

Minor issues

The authors confirm the elevator mechanism for SLC4 transporters. This makes the time right to move away from the core/gate nomenclature historically established by Lu et al. 2011 and switch to the transport/scaffold nomenclature commonly used for elevator transporters as suggested by Ficici et al. 2017 (ref 37).

We thank the reviewer and agree with the points raised. We agree that the gate/core nomenclature used in the literature is confusing since its original usage with regards to the UraA transporter. However, given that essentially all previous SLC4 publications have used the older nomenclature, we have decided to still refer to the original terminology to prevent confusion. In light of this and the points raised by both reviewers 2 and 3, we have added a footnote that addresses the issues with the usage of gate/core nomenclature.

The overall quality of the figures is below par. E.g., fig3: the superimposition makes the structures difficult to analyze (like suppl fig3 and suppl fig4), panel b has too much detail (like fig5).

Figs. 3b and 5b, and Supplementary Figs. 3 and 4 have been modified as per the reviewer's recommendations.

Line 84: indicate the protein was reconstituted in amphipols.

As indicated in the Methods section, we referred to the use of amphipol. We have not repeated this statement in the Results section.

Line 131: SLC29 should be SLC23

The change has been made.

Line 161: "exposes", please comment whether the unfolding of the beta-sheet is absolutely required for exposing R748 and E699. It appears that in the SLC26 and SLC23 IF structures changes in secondary structure are not required to expose the substrate binding site.

At this point, we cannot confirm with certainty if unfolding of the beta-hairpin is absolutely required for exposing R748 and E699. It is likely required for the elongation of TM11. It is possible that the whole beta-hairpin moves away through the cytoplasm to open water and ion access toward the IF cavity, however downward elevator motion of TM10 will be necessary for actual opening of the IF cavity. We did not observe such movement of the beta hairpin during the 1 microsecond MD simulations.

Line 170: this section shows large overlap with the previous section (line 147). I suggest the authors to combine these sections.

The modifications have been made as suggested.

Line 182: indicate whether the TM11 unfolding is observed in the MD simulations.

We did not observe unfolding of TM11 in the IF monomers during the 1 μ s MD simulations. We observed bending of TM11 in a few of the IF monomers, which might be a precursor for the unfolding. We have indicated this in the revised manuscript (page 12 in the revised manuscript): Alternatively, in the IF to OF transition, sideways motion of H1 toward TM11 may induce bending of TM11 (Supplementary Movies 4,5), leading to an occluded-like state, which precedes the α -helical unfolding and the consecutive β -hairpin formation (Fig. 6). We did not observe unfolding of TM11 in the IF monomers during the 1 μ s MD simulations.

Line 339 "elect", effect?

The change has been made.

Fig 3C legend: please remove "atomic". A surface representation can never be "atomic"

The change has been made.

Suppl fig7: please indicate what the densities mean, e.g., can this be related to a certain concentration and if so, are the contour levels identical for bicarbonate and chloride?

As suggested, the contour levels for the ion densities have been added in Supplementary Fig. 7 and other relevant figures.

REVIEWERS' COMMENTS:

Reviewer #1 (Remarks to the Author):

The authors have addressed the comments made in the original review, resulting in an improved version.

Reviewer #2 (Remarks to the Author):

The authors have done a good job in addressing my comments. Movie 1 provides a much clearer idea of the mechanism and the addition of a footnote to help to non-specialists navigate the manuscript is very welcome. I would be happy to see this work published.

Reviewer #3 (Remarks to the Author):

The authors have addressed my points appropriately.

I have one point concerning the following formulation:

Line 367 "ion gradient drives this ion toward the cytoplasm" >> in secondary transporters it is the sum of the gradients that drives transport indeed. However, on a molecular level the movement of the core domain is not 'driven' by a concentration gradient. After all, it has no ability to sense a concentration gradient. As the authors are aware, what in fact happens is that upon ligand binding the core domain gains the ability to reorient (in obligatory exchangers), and that the degree of being in this liganded, reorientation-competent state depends on the concentrations of the respective ligands. I suggest changing this sentence to read "In the OF state, the binding of the substrate ion at the ion coordination site allows the transformation of the OF to the IF state, which..". This is a relevant concept that readers new to the field may misinterpret with the current formulation.

Minor

Line 253 "Fecici" >> Ficici